# Emergence of social cluster by collective pairwise encounters in *Drosophila*

Lifen Jiang[1,2], Yaxin Cheng[2,3], Shan Gao[2,3], Yincheng Zhong[2,3], Chengrui Ma[2,3], Tianyu Wang[2,3], Yan Zhu[2,3,4]*

[1]School of Life Science, University of Science and Technology of China, Hefei, China; [2]State Key Laboratory of Brain and Cognitive Science, Institute of Biophysics, Chinese Academy of Sciences, Beijing, China; [3]University of Chinese Academy of Sciences, Beijing, China; [4]Advanced Innovation Center for Human Brain Protection, Capital Medical University, Beijing, China

**Abstract** Many animals exhibit an astonishing ability to form groups of large numbers of individuals. The dynamic properties of such groups have been the subject of intensive investigation. The actual grouping processes and underlying neural mechanisms, however, remain elusive. Here, we established a social clustering paradigm in *Drosophila* to investigate the principles governing social group formation. Fruit flies spontaneously assembled into a stable cluster mimicking a distributed network. Social clustering was exhibited as a highly dynamic process including all individuals, which participated in stochastic pair-wise encounters mediated by appendage touches. Depriving sensory inputs resulted in abnormal encounter responses and a high failure rate of cluster formation. Furthermore, the social distance of the emergent network was regulated by *ppk*-specific neurons, which were activated by contact-dependent social grouping. Taken together, these findings revealed the development of an orderly social structure from initially unorganised individuals via collective actions.

*For correspondence:
zhuyan@ibp.ac.cn

Competing interests: The authors declare that no competing interests exist.

## Introduction

Throughout the animal kingdom, collective behaviours are commonly observed at every scale, ranging from insect swarming to wildebeest migration (*de Bono and Bargmann, 1998*; *Mogilner et al., 2003*; *Ben Jacob et al., 2004*; *Buhl et al., 2006*; *Bazazi et al., 2008*; *Deisboeck and Couzin, 2009*). Assembling into social groups is a crucial survival strategy for many animals. For example, social groups are required for cooperative foraging (*Dombrovski et al., 2017*; *Dombrovski et al., 2019*), coordinating individuals' actions and enhancing vigilance (*Couzin, 2009*; *Nagy et al., 2010*; *Moussaïd et al., 2011*; *Jolles et al., 2017*).

Numerous investigations have attempted to quantify the collective behaviours of animal groups, often by extracting high-level features of massive structures emerging from seemingly stochastic actions of individuals (*Buhl et al., 2006*; *Nagy et al., 2010*; *Kim et al., 2012*; *Schneider et al., 2012*; *Ramdya et al., 2015*; *Jolles et al., 2017*; *Dombrovski et al., 2017*; *Jezovit et al., 2017*). One shared feature of collective behaviours is the maintenance of a certain distance, or social space, between nearby individuals. For example, individuals within a group of tufted ducks (*Conder., 1949*), cliff swallows (*Hutton, 1978*) or sheep (*King et al., 2012*) maintain particular distances from each other within a group, which are also dynamically modulated by environmental stimuli (*Bertrand et al., 2004*). However, it remains unclear how individuals act together to form a cohesive social group and how social distance is regulated or maintained.

The transmission and perception of social signals within a group also plays an essential role in collective behaviour, particularly in coordinating group-level actions. In locusts, mechanosensation of abdominal tactile stimuli promotes the coordination of mass migration, while vision is required to

detect the approach of others from behind (*Buhl et al., 2006*; *Bazazi et al., 2008*). In ants, tactile and chemosensory signals play an important role in cooperative foraging (*Billen, 2006*). Moreover, when traveling in a hierarchical group, pigeons rely on visual input from the left eye to respond quickly to the manoeuvres of others (*Nagy et al., 2010*). However, a systematic analysis of major sensory modalities to elucidate the functions underlying the assembly and sustainment of a social group has not yet been undertaken, partially due to a lack of suitable model organisms allowing for the effective manipulation of all members in a group with high precision and reproducibility.

*Drosophila melanogaster* is often overlooked when it comes to investigating group-level behaviours, despite the observation that fruit flies do indeed aggregate (*Guo et al., 2017*). In addition to assembling on food and oviposition sites (*Lihoreau et al., 2016*; *Dombrovski et al., 2017*), flies also exhibit local congregation without external artificial stimuli (*Stalker and Harrison, 1961*; *Navarro and del Solar, 1975*; *Simon et al., 2012*; *Burg et al., 2013*). The existing group-based behaviours, together with carefully controlled experimental conditions and powerful genetic tools, make fruit flies a promising neuroethological model for studying the mechanisms of collective behaviours (*Ramdya et al., 2015*; *Ramdya et al., 2017*; *Dombrovski et al., 2017*).

There are two major approaches among investigations of collective behaviours in *Drosophila*. One approach focuses on measuring the static end result, the last 'snapshot', of social aggregation and identifying potential sensory inputs (*Stalker and Harrison, 1961*; *Navarro and del Solar, 1975*; *Simon et al., 2012*; *Burg et al., 2013*), but does not examine the grouping process and its dynamics. The other approach is engaged in the study of patterns of interactions between individuals in a confined space, using networks of social interactions to understand information transmission (*Schneider et al., 2012*), but whether or how those described network properties contribute to social aggregation is not immediately clear. Interestingly, a study of the panic reactions of a group of flies to an aversive odour revealed features of collective avoidance (*Ramdya et al., 2015*). Social aggregation, in which individuals come together to eventually form a group, is distinct in many aspects from panic escape, where members of an existing group scatter. To date, there is no mechanistic framework explaining how dynamic social interactions between individuals (as a process) lead to a stable social aggregation (as a result), particularly in the absence of external stimuli.

In the current study, we established a paradigm for investigating connections between the collective actions of flies en masse and the process of social clustering, in which flies form a network with regularity. Analysis of the interaction dynamics of individual flies revealed that they rely on multiple sensory inputs, while collective local interactions drive the initiation and development of a stable, well-structured social cluster.

## Results

### Spontaneous formation of social clusters

Although previous work has focused on social spacing (*Simon et al., 2012*; *Burg et al., 2013*) and modelling social interaction networks (SINs) (*Schneider et al., 2012*) in *Drosophila*, the mechanisms by which social clusters are formed remain unclear. To clarify this question, we developed an improved behavioural paradigm enabling the distribution of flies in a two-dimensional space to be determined using a programmable 8-megapixel digital camera (*Figure 1—figure supplement 1A*). In our set-up, a group of 50 male or female flies was allowed to walk freely on the surface of an agar pad (1% agar) in a horizontally placed circular arena (diameter: 90 mm) without external stimuli. The agar pad provided a moist environment for prolonged observation, and a soft substrate for flies to walk on, thus constituting a controlled but relatively naturalistic setting associated with minimal stress or disturbance to the flies (*Figure 1—figure supplement 1A*).

After a short exploration period, a group of wild-type flies (*Canton-S, CS*) spontaneously aggregated toward a small region of the arena (*Figure 1A–C*; *Figure 1—figure supplement 1B,C*; *Figure 1—video 1*). Each fly was fitted to an ellipse to obtain its geometric parameters including length, width, orientation and centre of mass (*Figure 1B,C*). In subsequent quantifications, the location of a fly was represented by its centre of mass, and the distance between the centres of two flies was designated the inter-fly distance. To quantify the spatial distribution pattern of these flies, we first determined the accumulative distribution of all surrounding flies of each individual (designated as the reference fly) of one arena by aligning the centres of these individuals to the origin point and

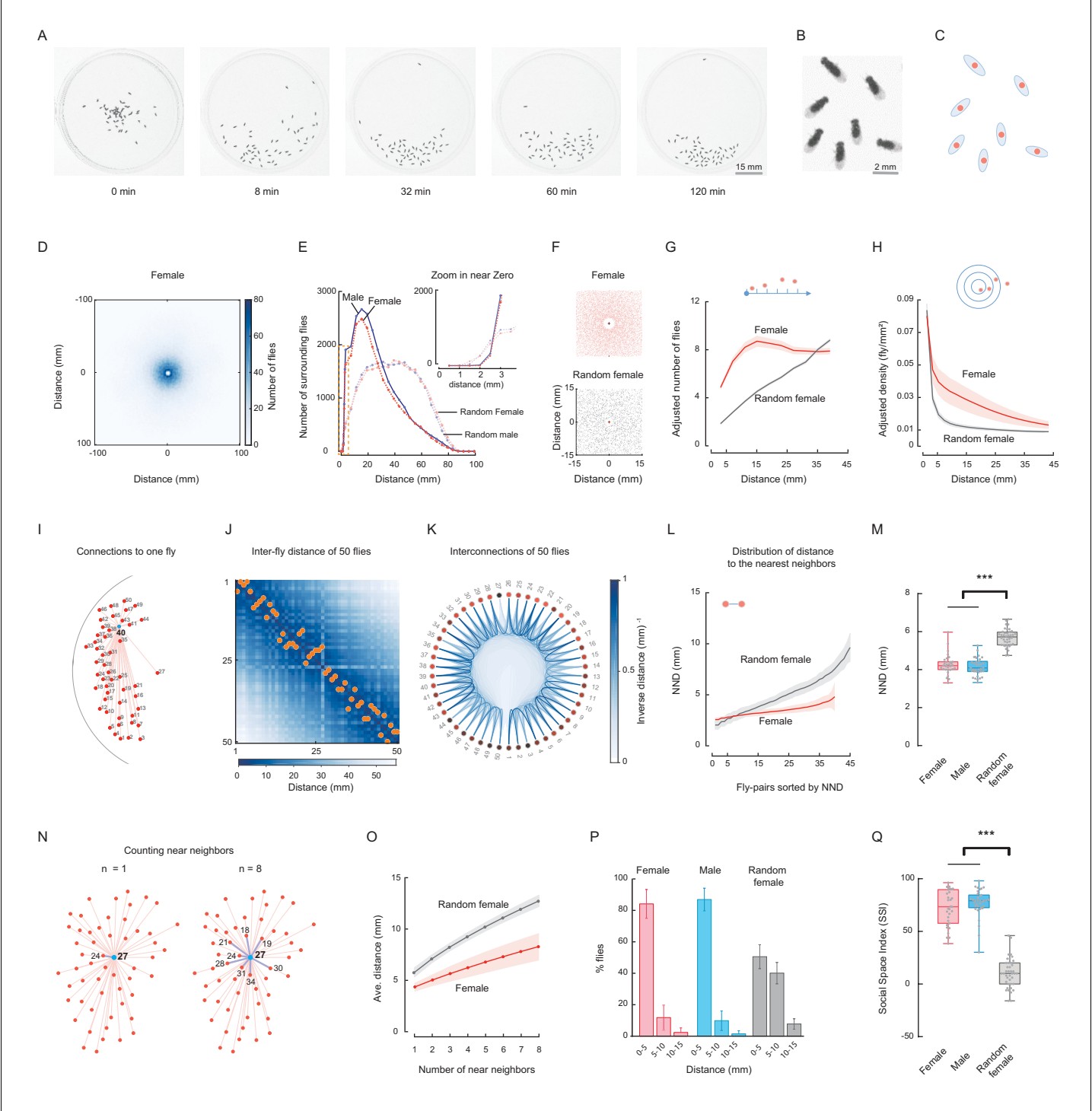

**Figure 1.** Spontaneous clustering of wild-type flies exhibits distinct spatial features. (**A**) Representative images show the distribution of a group of *Canton-S* (CS) male flies at the indicated time points. *Figure 1—video 1*. (**B**) Enlarged view showing the distribution of six flies in the last image of (**A**). (**C**) Representation of flies in (**B**) by their centres of body mass. (**D**) Image showing the merged distribution of all surrounding flies in 31 arenas with female *CS* flies. The origin was the aligned centres of each reference fly. *Figure 1—video 2*. (**E**) Quantification of distributions of merged surrounding flies at a distance from the centre. Dark red: *CS* female; dark blue: *CS* male; light red: random female (RF); light blue: random male (RM). N = 31, 35, 44 arenas. The insert plot on the left shows an enlarged view of the region near zero. (**F**) Images showing the enlarged views of the merged distribution of surrounding flies near the origin in wild-type female flies (up) and random female flies (bottom). N = 10 arenas. (**G**) Distributions of the area-adjusted number of surrounding flies over distance in female (red) and random female flies (grey). The bold lines indicate the average values over all arenas of the same type of fly, with the shaded areas indicate values within one s.e.m (N = 31, 40). (**H**) Distributions of the density of surrounding flies over

*Figure 1 continued on next page*

*Figure 1 continued*

distance in female and random female flies. The bold lines indicate the averaged values over all arenas for the same types of fly, with the shaded areas indicate values within one standard deviation (N = 31, 40). (I) Illustration of the measurements of all possible distances from one fly (#40) to others in the arena. Red dots represent individual flies, and numbers indicate their IDs. (J) Matrix of inter-fly distances between all 50 flies in the fly group in (I). The colour bar indicates the distance values. Orange dots marked the positions of shortest distance along each column, resulting in the NND and corresponding nearest neighbour of each fly on the bottom. All flies were females. (K) Circular representation of distance relationship between all 50 flies in (J). The intensities of the blue arcs connecting two flies correspond to the inverse distances between them. (L) Distributions of sorted NNDs of female and random female flies. Bold lines indicate the averaged distribution curve of sorted NNDs over all arenas, with the shaded areas indicate values within one standard deviation. N = 31, 40 arenas. (M) Distribution of the averaged NNDs of all flies in an arena. The flies were female (red), male (blue) and random female flies (grey), in 31, 35 and 40 arenas, respectively. (N) Illustration the first (left) and up to eighth (right) nearest neighbours of the designated reference fly (#27) in a group. Numbers indicate the fly IDs of the identified near neighbours. (O) Distribution of the mean multi-neighbour distances over the number of near neighbours. First the averaged n-near neighbour distance of all flies in an arena was calculated, then the distance values were averaged over all arenas (bold lines). The shaded areas indicate values within one standard deviation. Red indicates female flies and grey indicates random female flies. N = 31 and 40 arenas. (P) Histogram of flies with NNDs in the indicated ranges of distance, with bin 1 = 0–5 mm, bin 2 = 5–10 mm, and bin 3 = 10–15 mm. The types of flies were female (red), male (blue) and random female (grey). N = 31, 35 and 40 arenas, respectively. (Q) Social Space Index was calculated from (P) by subtracting the value of bin2 from that of bin1 in each arena. N = 31, 35, 40 arenas for female (red), male (blue) and random female (grey), respectively. In a box and whisker plot, the scatter points show all data points, the box includes the 25th to 75th percentile, the whiskers mark minimum and maximum, and the middle line indicates the median of the data set. ***: p<0.001 (one-way ANOVA followed with Tukey's *post hoc* test for multiple comparisons).

The online version of this article includes the following video, source data, and figure supplement(s) for figure 1:

**Source data 1.** Figure 1M, P-Q source data and related summary statistics.
**Figure supplement 1.** Experimental setup.
**Figure supplement 2.** Merged distribution of surrounding flies.
**Figure supplement 2—source data 1.** Figure 1—figure supplement 2 source data related summary statistics.
**Figure supplement 3.** Quantification of area-adjusted number of surrounding flies and density of surrounding flies.
**Figure supplement 3—source data 1.** Figure 1—figure supplement 3 source data related summary statistics.
**Figure supplement 4.** Estimating inter-fly connections via distance matrices.
**Figure supplement 4—source data 1.** Figure 1—figure supplement 4 source data.
**Figure supplement 5.** Analysing the distribution of Nearest Neighbour Distance.
**Figure supplement 5—source data 1.** Figure 1—figure supplement 5A-E source data related summary statistics.
**Figure supplement 6.** Characterisation of averaged distance to multiple near neighbours.
**Figure supplement 6—source data 1.** Figure 1—figure supplement 6 source data related summary statistics.
**Figure supplement 7.** Distribution of distance to the 1st nearest neighbours in wild-type flies and simulated flies.
**Figure supplement 7—source data 1.** Figure 1—figure supplement 7E source data and related summary statistics.
**Figure 1—video 1.** supplement to *Figure 1*.
https://elifesciences.org/articles/51921#fig1video1
**Figure 1—video 2.** supplement to *Figure 1D*.
https://elifesciences.org/articles/51921#fig1video2

---

superimposing the surrounding flies onto a plane (*Figure 1—video 2*). The superimposed patterns of all arenas with the same experimental conditions were then merged again (*Figure 1D*; *Figure 1—figure supplement 2A–D*). In distribution plots of the number of merged surrounding flies over distances, we found more flies in proximity to the origin than the outside, suggesting a strong tendency for aggregation (*Figure 1D,E*; *Figure 1—figure supplement 2A–D*).

We next addressed whether such aggregation arose just by chance when individuals walked independently in the arena. We compared the spatial distributions of these social flies with that of 'random flies', which were presumed to act independently, with no social interactions between them (see Materials and methods). As shown in *Figure 1E* and *Figure 1—figure supplement 2E-H*, random flies exhibited different spatial distributions of merged surrounding flies, with more flies dispersed away from the origin, suggesting a dependency between individuals of wild-type groups to form spatial patterns, which was absent in a random distribution. Closely examining the distributions near the origin revealed that wild-type flies maintained a longer distance from their nearest neighbours than random flies (*Figure 1F*). Thus, wild-type flies maintained a larger impermissible zone regarding social space.

We then calculated the distribution of surrounding flies using area-adjustment. As shown in *Figure 1—figure supplement 3A*, because flies were located only inside the circular arena, directly

counting the number of surrounding flies may cause biased results, as the measuring distance (from the centre of each reference fly) increased beyond the arena edge. To compensate for under-estimation of the number of surrounding flies, we made an adjustment with the actual area within the arena (the intersecting area between the circle of the arena and the circle with a radius of the measured distance and a centre at the reference fly) (*Figure 1—figure supplement 3A*). As shown in *Figure 1G* and *Figure 1—figure supplement 3B and C*, compared with random flies, the wild-type flies had a greater number of surrounding flies over most of the measured distances. In other words, at a distance from any individual, there were, on average, more surrounding flies in a group of wild-type flies, suggesting a strong social aggregation of wild-type flies. Similarly, we calculated the density of surrounding flies within a distance. In wild-type flies, the density of surrounding flies decreased gradually as the measuring distance increased, whereas the density in random flies was lower and decreased faster (*Figure 1H*; *Figure 1—figure supplement 3D,E*).

Besides quantifying the number of surrounding flies, we analysed the distance properties between the neighbouring flies. First, we calculated the distance matrix describing the distances between all possible pairs of flies in the group (*Figure 1I,J*). The distance matrix enabled us to identify the nearest neighbour of each fly (*Figure 1J*; *Figure 1—figure supplement 4A,B,D,E*) and to visualise inter-fly spatial nearness at the group level (*Figure 1K*; *Figure 1—figure supplement 4C, F*). Apparently, in wild-type flies, there were more near neighbours with similar short inter-fly distance than in random females (*Figure 1K*; *Figure 1—figure supplement 4C,F*), which was further confirmed when analysing the distribution of the nearest neighbour distance (NND) of all flies in an arena (*Figure 1L*; *Figure 1—figure supplement 5*). In contrast, the inter-fly distance of random flies began low but quickly exceeded that of wild-type flies when more additional pairs were considered (*Figure 1L*; *Figure 1—figure supplement 5A–E*). The average NND over all arenas also indicated that wild-type flies in a group had a smaller social distance on average than random flies (*Figure 1M*).

We set to quantify the nearness of a fly to its neighbours by measuring the average distance of a fly to its multiple near neighbours (up to 8-th nearest neighbours, including the nearest neighbours, *Figure 1N*; *Figure 1—figure supplement 6*). For both wild-type flies and random flies, the averaged multi-neighbour distance increased as more near neighbours were included (*Figure 1O*; *Figure 1—figure supplement 6T*), however, multi-neighbour distances in wild-type flies were smaller and increased more slowly than those of random flies (*Figure 1O*; *Figure 1—figure supplement 6T*). As the trend was consistent for the numbers of near neighbours from 1 to 8, we used the distance to the 1st nearest neighbour (equivalent to NND) as the basis for further quantifying the fly group (*Figure 1—figure supplement 7A–D*).

NND has been associated with social space (*Mogilner et al., 2003*; *Simon et al., 2012*). Wild-type flies displayed surprisingly consistent population NND for up to 50 min, and exhibited much lower NND than random flies (*Figure 1M*; *Figure 1—figure supplement 5E*). Consistent with previous observations (*Simon et al., 2012*; *Navarro and del Solar, 1975*), a large proportion of wild-type flies (84 ± 9% for females, 87 ± 7% for males) exhibited a social distance within 5 mm from each other, approximately 1.5–2 body lengths (*Figure 1P*; *Figure 1—figure supplement 7D*), while a significantly lower proportion (10%–11%) of wild-type flies lay within 5–10 mm of their nearest neighbour. Conversely, random flies exhibited a lower proportion within 0–5 mm (51 ± 8%) but a higher proportion within 5–10 mm (40 ± 7%) compared with wild-type flies.

We employed the 'Social Space Index' (SSI), based on the NND, to quantify distribution en masse, as described previously (*Simon et al., 2012*). A histogram of NND values for all flies in an arena was built using bins of 5 mm increments (*Figure 1P*). SSI values were calculated as the percentage of flies in the first bin (NND range: 0–5 mm) minus that in the second bin (NND range: 5–10 mm) (*Figure 1P,Q*); a larger SSI value indicates a smaller inter-fly distance. Our results revealed that wild-type fly groups exhibited significantly higher SSI values than the random flies (72%–77% vs 10%), in accord with previous studies (*Figure 1Q*) (*Burg et al., 2013*; *Simon et al., 2012*).

Our comparison between the empirical results of wild-type flies with simulations of random flies suggested that self-organised clusters of wild-type flies would not emerge without interactions between individuals.

## Social clusters in fruit flies are well-structured networks

To elucidate the local relationships of clustered flies, we designed a six-step procedure to define a cluster and further quantify the local regularity of the resultant cluster (*Figure 2A*; *Figure 2—video 1*). Our algorithm used two criteria to build a cluster: area threshold and distance threshold. The algorithm first identified *and* grouped flies with residing areas smaller than an area threshold into a basic cluster, then repeatedly incorporated nearby flies within a threshold distance (*Figure 2A*). Due to the inherent uncertainty of determining whether an object belongs to a cluster, we evaluated five *criterial sets for clustering* (CSC), with stringency from high to low (*Figure 2B*, see Materials and methods). Higher stringency limited the number of qualified flies, and resulted in smaller clusters or no clusters at all (*Figure 2—figure supplement 1*).

With each set of criteria, clusters in all arenas were automatically identified, and the percentage of arenas with clusters was then calculated for three types of flies: male, female, and random female (*Figure 2C*). Furthermore, the percentage of the number of clustered flies, out of the total flies in an arena (designated as CT value), was calculated then averaged over all arenas. As shown in *Figure 2—figure supplement 2A,B*, starting with criterial set #3, random flies began to show substantial cluster contents. Overall, male and female wild-type flies exhibited similar tendencies of increasing cluster size as the criteria became less stringent. The CT values of random flies also increased, but were far smaller than those of wild-type flies (*Figure 2—figure supplement 2B*). The maximal difference between wild-type flies and random flies occurred when using criterial set #2, by which only a smaller number of random flies were close enough to form clusters (*Figure 2D*; *Figure 2—figure supplement 2B*).

Based on their locations in a cluster, flies were further classified into outsiders (the flies constituting the periphery of a cluster) and insiders (those being inside of a cluster) (*Figure 2A*, Step 5). Plotting the percentage of the number of insiders out of the total flies in an arena (designated IT value), revealed tendencies similar to the CT values in female, male and random female flies (*Figure 2E*; *Figure 2—figure supplement 2C,D*). Furthermore, when comparing the composition of clusters in three types of flies, the clusters formed by wild-type flies had more insiders than random flies (*Figure 2—figure supplement 2E,F*). Therefore, even when random flies formed clusters under relaxed criteria, the organisation of such clusters still differed from that of wild-type flies (*Figure 2—figure supplement 1A*–C; *Figure 2—figure supplement 2E–F*), suggesting that the formation of clusters by wild-type flies minimises the perimeter of the cluster, thus displaying higher efficiency in packing the cluster, while still maintaining impermissible distances between the members (*Figure 1—figure supplement 5E*).

The residing area of individuals, particularly the insiders, was an important parameter describing the local properties of a cluster. We found that as the criteria became relaxed, the residing area increased greatly in random flies, but increased only slightly in female and male flies (*Figure 2—figure supplement 3A–E*), indicating that different settings of cluster-defining criteria did not significantly influence the cluster properties under consideration.

The social clusters of wild-type flies exhibited unique structural features of a typical distributed network (*Baran, 1964*), with individual flies as the interlinked nodes (*Figure 2A*, Step 6). Thus, the flies self-organised into a structure with nearly uniform near-neighbour connections, similar to the lattice arrangement of atoms in a crystal. The number of links of a fly to its contiguous neighbours, a measure of connectivity (*Figure 2A*, Step 6), was $5.53 \pm 0.19$ for males and $5.49 \pm 0.30$ for females when evaluated with criterial set #2 (*Figure 2F*; *Figure 2—figure supplement 3F*). Furthermore, the average distance between these contiguous neighbours (the length of links) was $4.70 \pm 0.36$ for males, $4.65 \pm 0.41$ for females and $4.27 \pm 1.13$ for random flies (*Figure 2G*; *Figure 2—figure supplement 3G*).

We noticed the near-uniform distributions of the flies in arenas and proceed to quantify the variation in the inter-fly spacing. To simplify the analysis, we considered the variation in the *distance of contiguous neighbours* (DCN) of inner flies, who did not border the arena edge (*Figure 2—figure supplement 4A*). The distribution of standard deviations of DCN in wild-type flies displayed a narrower peak and a smaller median ($1.66 \sim 1.76$ mm) than that in random flies (median: 4.07 mm) (*Figure 2—figure supplement 4B*). Furthermore, when evaluating an arena with the variation of DCN of only the inner flies with smaller standard deviations, which were likely associated with clustered flies, we found that the wild-type flies also exhibited smaller variations (*Figure 2—figure supplement*

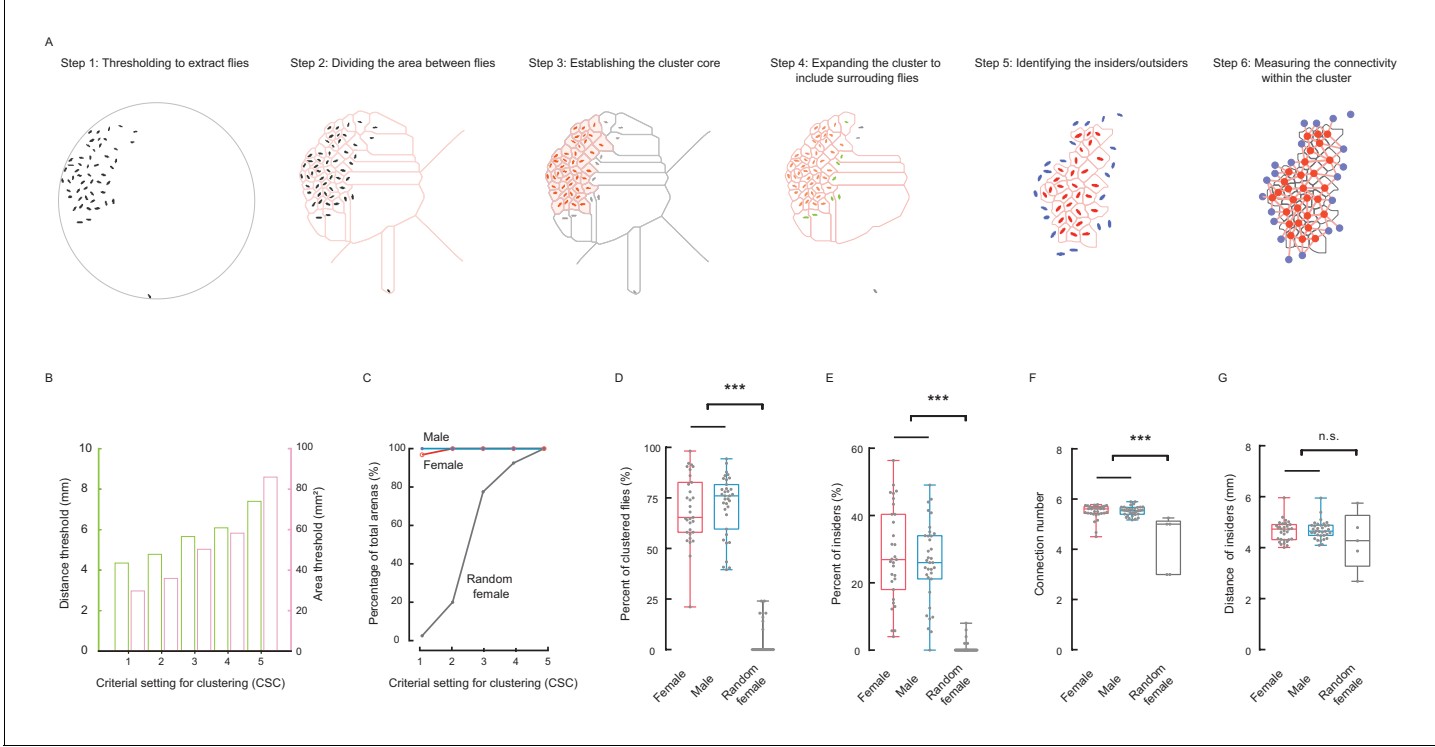

**Figure 2.** Social cluster represents a well-structured network. (**A**) A six-step procedure to automatically reconstruct a cluster from a group of flies in two dimensional space. Step 1: Using digital image processing methods to extract the pixels of each fly from a raw image and calculating the geometric properties of the pixel set of that fly. Step 2: Dividing the area between all flies and the edge of arena. The divided area surrounding each fly is designated as its residing area. Step 3: Establishing the basic cluster by combining flies whose residing areas are smaller than an area threshold. Each red ellipse indicates a fly incorporated by the cluster; the corresponding pink shaded area indicates its residing area. Step 4: Repeatedly expanding the cluster to include surrounding flies (green) within a threshold distance. The leftover fly was coloured grey. Step 5: In the resultant cluster, identifying the insiders (red) and outsiders (blue). Step 6: Quantifying the local regularity of the cluster. *Figure 2—video 1*. (**B**) Showing the area threshold and distance threshold (as a set) to reconstruct clusters in (**A**). Five criterial settings for clustering (CSC), with stringency from high to low, were defined and evaluated in the following panels. (**C**) Comparing the percentages of arenas formed a cluster under the CSC from 1 to 5. N = 31, 34, 40 arenas for female, male and random female flies, respectively. (**D**) Average percentage of clustered flies (CSC = 2). (**D**) Average percentage of clustered flies (of total flies in an arena) (CSC = 2). (**E**) Average percentage of insiders of total flies (CSC = 2). (**F**) Average number of connections from an insider to its contiguous neighbours (CSC = 2). (**G**) Average distance of an insider to its contiguous neighbours (CSC = 2). N = 31, 34, 40 arenas in (C–E) and N = 31, 34, five arenas in (F–G) for female, male and random female flies, respectively. Data in (**D, E**) were part of *Figure 2—figure supplement 2A and C*; Data in (**F,G**) were part of *Figure 2—figure supplement 3F and G*. In a box and whisker plot, scatter points show all data points, the box includes 25th to 75th percentile, the whiskers mark minimum and maximum, and the middle line indicates the median of the data set. ***: p<0.001 (one-way ANOVA followed with Tukey's post hoc test for multiple comparisons).

The online version of this article includes the following video, source data, and figure supplement(s) for figure 2:

**Source data 1.** Figure 2D-G source data and related summary statistics.
**Figure supplement 1.** Patterns of clusters reconstructed with different clustering criteria.
**Figure supplement 2.** Exploring the settings of cluster criteria on cluster reconstruction.
**Figure supplement 2—source data 1.** Figure 2—figure supplement 2 source data and related summary statistics.
**Figure supplement 3.** Characterising the properties of identified clusters.
**Figure supplement 3—source data 1.** Figure 2—figure supplement 3 source data and related summary statistics.
**Figure supplement 4.** Analysing the variation within distances of the contacting neighbours.
**Figure supplement 4—source data 1.** Figure 2—figure supplement 4 source data and related summary statistics.
**Figure supplement 5.** Social clustering is independent of arena size.
**Figure supplement 5—source data 1.** Figure 2—figure supplement 5B-C source data and related summary statistics..
**Figure supplement 6.** Comparing spatial patterns of random dots with that of real flies.
**Figure supplement 6—source data 1.** Figure 2—figure supplement 6 source data and related summary statistics.
**Figure supplement 7.** A simple model of clustering by the combined effects of global attraction and local repulsion in a two-dimensional space.
**Figure supplement 8.** Social experience, physiological state, age and circadian rhythm modulate social clustering in wild-type flies.
**Figure supplement 8—source data 1.** Figure 2—figure supplement 8A-D source data and related summary statistics.
**Figure supplement 9.** Grooming did not affect social clustering.
*Figure 2 continued on next page*

*Figure 2 continued*

**Figure supplement 9—source data 1.** Figure 2—figure supplement 9C-D source data and related summary statistics.
**Figure 2—video 1.** supplement to *Figure 2A*.
https://elifesciences.org/articles/51921#fig2video1

*4C*). As a small variation of a dataset is indicative of the homogenous nature of the measured property, the similar distances between contacting neighbours reflected a regular organisation of local near-neighbours across the entire social network of wild-type flies. The regularity of local spacing in the social clusters implied that cluster formation possibly involves cascades of local interactions guided by a set of common principles.

Behaviours of *Drosophila* in general are reported to be regulated by external (environmental) cues, internal (physiological) states and social experiences (*Lihoreau et al., 2016*; *Ramdya et al., 2015*; *Kent et al., 2008*; *Krupp et al., 2008*; *Battesti et al., 2012*; *Levine et al., 2002*; *Guo et al., 2017*). As shown in *Figures 1* and *2*, both male and female wild-type flies exhibited similar abilities to form clusters.

Quantification of cluster formation of 50 flies in circular arenas of different diameters (ranging from 90 mm to 170 mm) demonstrated that cluster formation was independent of the diameter or area of the testing arena, whereas the spatial features of random distribution were severely influenced by the size of arena (*Figure 2—figure supplement 5A,B*). Additionally, our results revealed that clusters were readily formed only when the number of testing flies was greater than 10 (*Figure 2—figure supplement 5C*).

We next surveyed the impacts of various factors on social clustering in wild-type flies. To better understand the relationship between the size of arena, the number of flies and social distance, we established groups of 'random dots' with random locations, which are set a minimal distance (impermissible distance or hard-core distance, see Materials and methods) away from each other (*Figure 2—figure supplement 6A-F*) (*Baddeley et al., 2015*). Average NND decreased as the population increased in random dots (*Figure 2—figure supplement 6E,F*) and in actual flies (*Figure 2—figure supplement 6G*). These NND hard-core distance curves also revealed that, in random dots, the NNDs of populations with long hard-core distances were also greater than those of NNDs in populations with short distances (*Figure 2—figure supplement 6E,F*). Importantly, the NND-hard-core distance curve of wild-type flies exhibited even lower values than that of random dots with 0 minimal distance (*Figure 2—figure supplement 6G*), suggesting that social clusters of actual flies, even with average minimal contact distances of approximately 2.5 mm (*Figure 2—figure supplement 6E*), behave in opposition to the Matérn hard-core point process (*Baddeley et al., 2015*). As a hard-core distance is indicative of local repulsion, tightness of clusters by actual flies suggested a global force of attraction on top of the local repulsion during cluster formation.

To evaluate whether the patterns of social clustering develop solely from interactions of attractive and repulsive cues without considering specific dynamic processes, we established a simplified model in which a fly walks under the combined influence of attraction and repulsion (*Figure 2—figure supplement 7*), based on three assumptions. First, every fly in the arena generates fields of attraction and repulsion with different distance profiles (*Figure 2—figure supplement 7A*). Second, the net result of these forces on a single fly can be calculated based on vectorial addition (*Figure 2—figure supplement 7B*). Third, a set of flies already establish a small cluster, which is relatively stable and not affected by an approaching fly (*Figure 2—figure supplement 7B,C*). Using Monte Carlo simulation, we found that a fly is strongly repelled by the fly set at close distances, but when the distance increases beyond a certain limit, the net force changes into attraction (*Figure 2—figure supplement 7C*). Even when the relative strength between repulsion and attraction (defined as the C factor) is adjusted over a broad range, such the abrupt switch from repulsion to attraction remains unchanged (*Figure 2—figure supplement 7D*). Importantly, the net force decreases quickly to a very low level as distance increases, but remains attractive. As a result, a fly walking in a force field generated by a set of relatively stable flies, - consistently exhibits a tendency to move toward the set, if it is far away, but eventually stops approaching when it reaches the interface of zero net force at a critical distance to the set (*Figure 2—figure supplement 7E*). The opposite forces on each side of the interface squeeze the fly to a defined distance from the set, and apparently to become a new

member of the set. Therefore, the antagonistic interaction between broad yet weak attractions and local but strong repulsions leads to a regular spacing between the clustered flies.

Furthermore, the SSI (Social Space Index) values of socially isolated flies (*Xie et al., 2018*; *Simon et al., 2012*), hungry flies and aged flies were lower than those of corresponding controls, suggesting a positive influence of social experience, and negative impacts of hunger and aging on clustering (*Figure 2—figure supplement 8A–C*). Notably, social clustering in females exhibited strong oscillations closely related to circadian rhythm (*Figure 2—figure supplement 8D*).

To evaluate the influence of self-grooming, we artificially increased grooming events by dusting flies with fine particles (*Seeds et al., 2014*) (*Figure 2—figure supplement 9A–C*). Compared with untreated controls, dusted flies exhibited increased grooming events, but no significant changes in SSI values (*Figure 2—figure supplement 9D*).

Overall, we demonstrated that fruit flies self-organised collectively into an orderly cluster with the topology of a distributed network. The robustness and unique features of such social clustering prompted us to further investigate the dynamic processes and mechanisms underlying cluster formation.

## Collective dyadic interactions contribute to the clustering process

To better understand the dynamic process of social clustering, we recorded the arena with a camcorder from above and analysed the performance of groups of flies from video sequences. Massive sporadic movements of flies resulted in location changes for every individual by the end of the process. Individual flies walked for various distances, sometimes exploring the arena for a long time before occupying a final position in the cluster. During the period, flies involved numerous interactions with other flies.

It took wild-type flies approximately 4 min (3.6 ± 0.3 min for female flies and 3.8 ± 0.2 min for male flies), 8–10 min (9.9 ± 0.8 min for female flies and 7.6 ± 0.2 min for male flies) and 22 min (22.2 ± 0.6 min for female flies and 21.9 ± 1.2 min for male flies) to reach cluster sizes of 10, 25 and 45 flies, respectively (*Figure 3A*). As the clusters grew quickly from 5 to 20 min, we subdivided this period into three 5 min phases (Stage 1, Stage 2 and Stage 3; *Figure 3A*).

Inter-fly interactions occurred exclusively between two flies, and the timing of these dyadic interactions appeared to be stochastic. We used two criteria to define an encounter event for a pair of flies: a) the distance between them was within 1.5 body length, and b) the approaching fly was facing and walking toward the other fly (i.e., the other fly was within its frontal 180° view). As shown in *Figure 3B*, numerous encountering events occurred during cluster formation with the overall number of encountering events in males (509 per group) being twice that in females (237 per group). Further, by surveying all dyadic interaction events, we calculated the frequency of inter-fly encounters at each phase across the entire arena. Interaction events were more frequent at first (100 ± 5 for females and 203 ± 9 for males in Stage 1) and then decreased over each consecutive 5 min (*Figure 3C*). These observations indicated that inter-fly interactions might contribute to the nucleation, growth and maturation of the social cluster.

## Asymmetric interactions and stereotypic consequences of pair-wise social encounters

Closer inspection revealed that, in wild-type flies, inter-fly encounters were mainly asymmetric, occurring by one walking fly actively approaching a stationary fly, similar to findings reported by *Schneider et al. (2012)*. The proportions of such asymmetric interactions among all encounter events in Stages 1–3 were 99%, 100%, and 100% in female groups, and 99%, 98%, and 98% in male groups, respectively. Adequate behavioural responses were elicited by the active fly through gentle touches of peripheral appendages when approaching the stationary fly (*Figure 3—video 1*).

To characterise the dyadic interactions for an encounter event, we defined the actively approaching fly as the 'interactor' and the stationary fly as the 'interactee' in our behavioural observation (*Figure 3D*). The encounters mostly (probability of 94%) led to active physical contact via legs and wings of both flies. The appendage-touch points were classified into eight types of actions from the point of view of the 'interactee', namely, frontal touch (F), rear touch (Rear or Wing) and leg touches (Legs: L1–L3 and R1–R3) (*Figure 3D*). With an interactive behavioural labelling program, we obtained details of encounter events, including appendage-touch points and responses of the pair

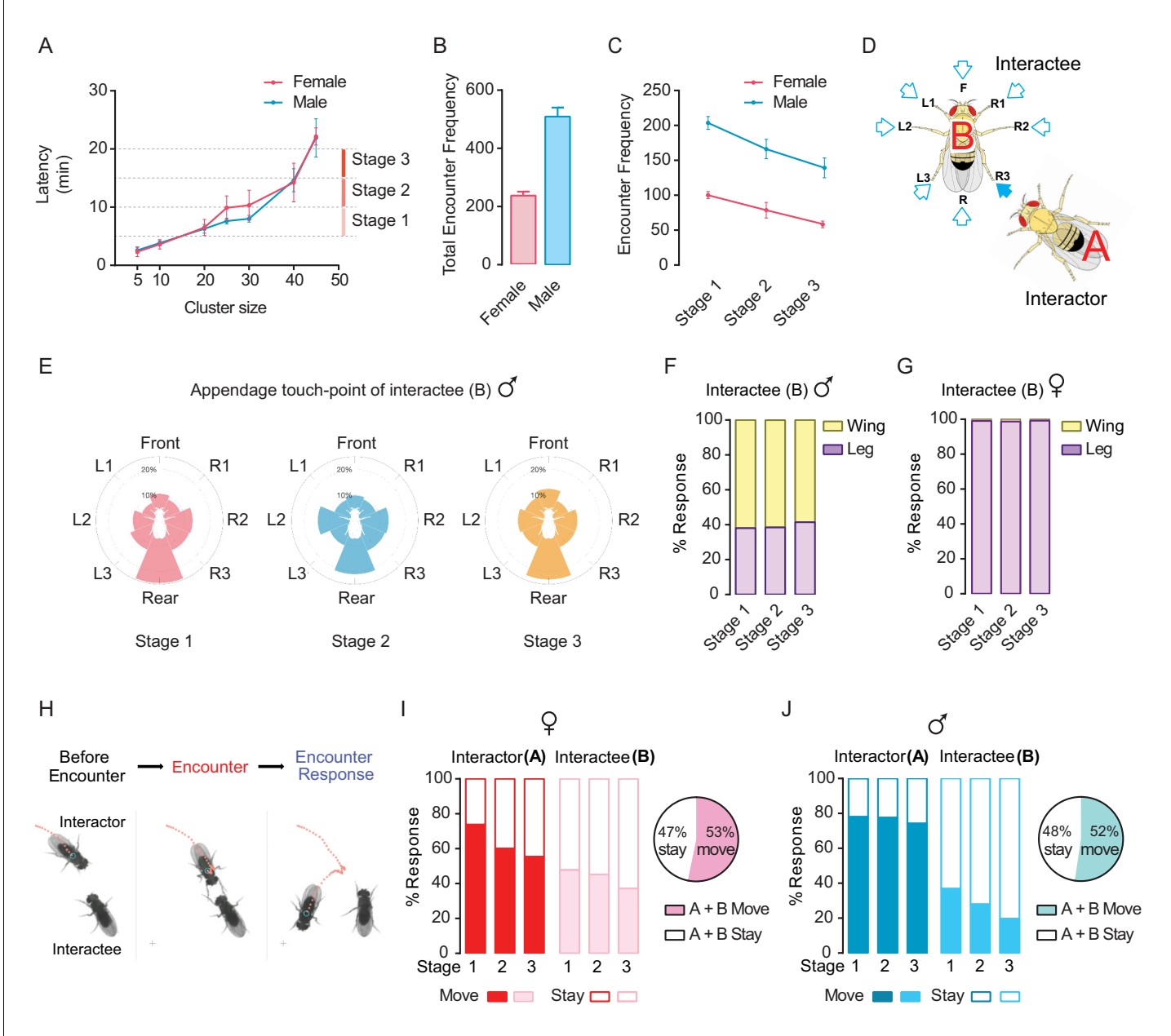

**Figure 3.** Dyadic encounter events are the main form of interactions during cluster formation in wild-type flies. (**A**) Rapid increase of the cluster size (number of flies) through incorporating more flies during clustering (N = 7 arenas). (**B**) Total encounter events during the period between 5 and 20 min (N = 5 arenas). (**C**) The number of encounter events in three stages of cluster growth (N = 5 arenas). (**D**) Schematic of an encounter event to show the appendage touch-points on the Interactee. Eight touch-points were defined: F (Frontal), L1-L3 and R1-R3 (Legs), R (rear or wings). (**E**) The proportion of body points touched by an 'Interactor' in male flies. Flies are shown in white at the centre and the proportion of each touch point is presented by the length of colour-coded bar. N = 5 arenas, and the number of encounter events = 203, 180, 140 for stages 1, 2 and 3. (**F, G**) Percentage of behavioural responses of the 'Interactee' after touching by the 'Interactor', in males (**F**) and females (**G**). The 'Interactee' used wings (Wing) or legs (Leg) to repel the 'Interactor' after being touched. N = 5 arenas, number of events = 471 (male) and 199 (female). (**H**) Three images from a video sequence showing behaviours by the 'Interactor' and 'Interactee' during an encounter event. The red dashed line indicates the locomotion trajectory of 'Interactor'. *Figure 3—video 1.* (**I, J**) Behaviour outputs after social encounters in the 'Interactor' and 'Interactee' in females (**I**) and males (**J**). Left panel: percentage of behavioural responses in the three stages; Right panel: percentage of net movement of encountered pairs (all three stages combined, quantified from the left panel). Stay: stay at the original location after encountering. Move: move away after encountering. N = 5 arenas, number of events = 1046 (male) and 474 (female).

The online version of this article includes the following video, source data, and figure supplement(s) for figure 3:

**Source data 1.** Figure 3A–C, E–G, and I–J source data and related summary statistics.

*Figure 3 continued on next page*

*Figure 3 continued*

**Figure supplement 1.** Quantification of touched sites and behavioural outputs of encounters in wild-type female flies.
**Figure supplement 1—source data 1.** Figure 3—figure supplement 1B source data.
**Figure supplement 2.** An image sequences of an encounter event in males.
**Figure 3—video 1.** supplement to *Figure 3*.
https://elifesciences.org/articles/51921#fig3video1

of flies after encountering, as in the example data from wild-type females shown in *Figure 3—figure supplement 1A* and *Figure 3—video 1*. The approaching flies preferred to use their forelegs. Furthermore, in male encounters, the frequencies of rear-touches on the 'interactee' were higher than that on other sites (*Figure 3E*), and the 'interactee' responded more frequently with wings (61%, percentage of all responses over three stages) than with legs (39%) (*Figure 3F*). In females, however, the 'interactor' was more likely to approach the 'interactee' from the side and behind (*Figure 3—figure supplement 1B*), and responses of the 'interactee' relied mainly on legs (99%) and rarely on wings (1%) (*Figure 3G*), revealing that female and male flies adopt different strategies for social encounters. These results suggest that sexually dimorphic neural mechanisms mediate social aggregation behaviour in flies.

We next examined the consequences of encounters in pairs of flies, focusing on changes of locomotion of the pair, which collectively contribute to the group dynamics. After encountering, one or both flies would subsequently move away, but also become stationary ('interactor') or remain stationary ('interactee') (*Figure 3H*; *Figure 3—figure supplement 2*; *Figure 3—video 1*). As shown in *Figure 3I and J*, female and male flies exhibited different tendencies for movement after encountering. In females, the 'interactor' tended to 'move-away' ('A Move', with alteration of walking trajectory) after encountering (*Figure 3I*). However, over time, in the growing and maturation phases, the proportions of 'interactors' that moved-away and stayed ('A Move' and 'A Stay') became similar (*Figure 3I*). For the female 'interactee', the tendencies to 'move-away' or 'stay' after encountering were approximately equal, with a slight increase of stay response over time (*Figure 3I*). In contrast, in male flies, a large proportion of 'interactors' preferred to 'move-away' rather than becoming stationary, while a higher proportion of 'interactees' remained stationary rather than walking away (*Figure 3J*). We observed an intriguing common pattern among males and females, by which, after a social encounter, the stationary fly tended to remain standing, whereas the incoming fly tended to move again and further 'probe' multiple stationary flies before settling down (*Figure 3I,J*). The scale and scope of social encounters and stereotypic 'stay-or-move' responses suggested that social encounters not only generated the group-level dynamics prerequisite for cluster formation, but also actively drove cluster development.

## Clustering grows by social encountering at the border

To understand how seemingly spontaneous pairwise interactions result in a structured social network, we examined the contribution of encounter responses to cluster development. We focused on encounter events occurring at the periphery of a cluster. When a moving fly reached the cluster, it was likely (97% in male, 77% in female) to interact with flies composing the border, who were stationary. These 'moving' and 'stationary' flies were classified into three categories according to their final decisions after an encounter: stay in place, move out of the cluster, and move into the cluster (*Figure 4A*; *Figure 4—video 1*). For both males and females, most encounters by a pair of 'moving' and 'stationary' flies resulted in either moving in or staying at the border of the cluster, thereby increasing the number of flies in the cluster (*Figure 4B–H*; *Figure 4—figure supplement 1A–G*).

Specifically, as shown in *Figure 4B–C*, the 'moving' males were inclined to 'Move-in' (76%) to join the cluster after an encounter, while a small proportion (3%) of 'moving' males directly walked into a cluster without any prior interactions with others at the border. Only 10% of 'moving' male flies that chose to walk away after encounters at the cluster edge (*Figure 4B,C*). The 'moving' female flies displayed a similar tendency to join the cluster, with only 3% of moving females leaving the cluster (*Figure 4—figure supplement 1A,B*). The 'stationary' flies, which stayed at the cluster edge before encounters, showed a high rate (89% in males and 67% in females) of remaining within the cluster after an encounter (*Figure 4D,E*; *Figure 4—figure supplement 1C,D*).

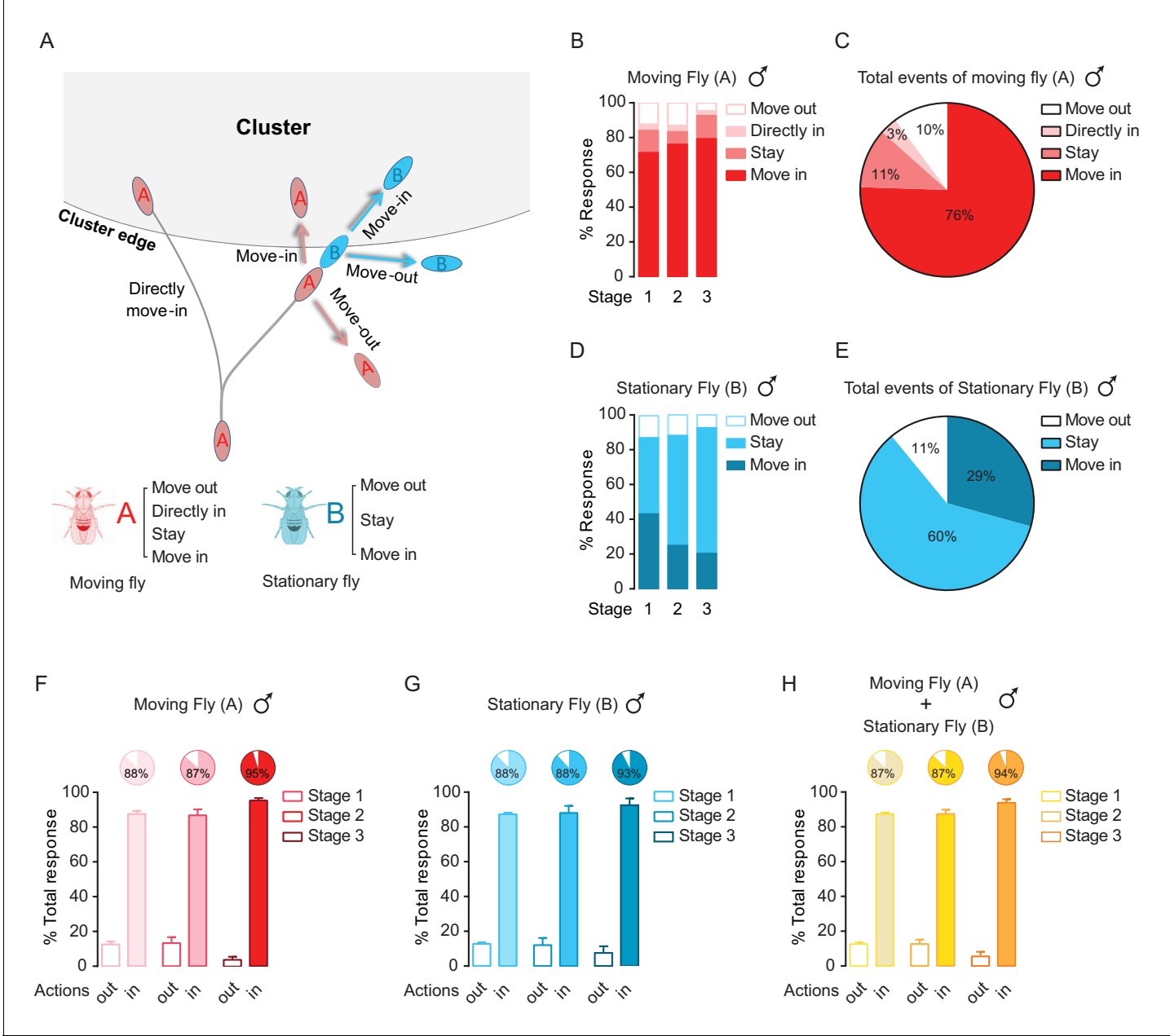

**Figure 4.** Dyadic interactions drive clusters to grow. (**A**) A schematic showing possible events occurring near the border of a cluster. Letter 'A' and 'B' are designated as the walking fly and the stationary fly (standing at the border of a cluster), respectively. 'Move out': move away to leave the cluster. 'Move in': move to join the cluster. 'Stay': stay near the cluster edge. 'Directly move in': the walking fly joins the cluster without first interacting with any flies. (**B**) Percentage of different encounter outputs of male walking flies after encounters at the cluster edge at the indicated stages. (N = 4 arenas, total number of encounter events = 92). (**C**) Total behavioural outputs of male walking flies after encountering at the cluster edge. Data were from (**B**). (**D**) Percentage of encounter responses of male stationary flies at the cluster edge after encountering (N = 4 arenas, total number of encounter events = 92). (**E**) Total behavioural outputs of male stationary flies at the cluster edge after encountering. Data were from (**D**). (**F, G**) The combined percentage of joining or leaving the cluster of walking flies (**F**) and stationary flies (**G**) after encountering at the cluster edge. Data sets came from (**C**) and (**E**), respectively. 'in' includes 'move in' + 'stay' + 'directly move in', 'out' is 'move out'. (**H**) Total percentage of behavioural output of the pairs after encountering, indicating the combined contributions by walking and stationary flies to cluster growth. Data were from (**F**) and (**G**). Values shown in (**F–H**) are mean ± s.e.m.

The online version of this article includes the following video, source data, and figure supplement(s) for figure 4:

**Source data 1.** Figure 4B, D, F, and G–H source data and related summary statistics.

**Figure supplement 1.** Quantifying behavioural outputs of female encountering at the cluster edge.

**Figure supplement 1—source data 1.** Figure 4—figure supplement 1A-G source data.

*Figure 4 continued on next page*

*Figure 4 continued*

**Figure supplement 2.** Characterising distributions of encounter events in typical regions of arena during clustering of wild-type female flies.
**Figure supplement 2—source data 1.** Figure 4—figure supplement 2B source data.
**Figure 4—video 1.** supplement to *Figure 4*.
https://elifesciences.org/articles/51921#fig4video1

The combined high rate of joining the cluster ('Move in' + 'Stay' + 'Directly move in') and low rate of leaving the cluster ('Move out') by both 'moving' and 'stationary' flies effectively contributed to cluster expansion in both males and females (*Figure 4F–H*; *Figure 4—figure supplement 1E–G*). Over 5 to 20 min (stage 1–3), almost every event at the cluster edge accounted for an increase in cluster size (*Figure 4H*; *Figure 4—figure supplement 1G*).

The final positions of the flies joining the cluster were also dynamically determined. From video sequences, we observed that a new fly usually walked inside the cluster, disturbing other flies along the way (*Figure 4—video 1*), before it finally stopped. As the number of members of a cluster increased, the number of inter-fly interactions inside the cluster also increased (*Figure 4—figure supplement 2*). These redistribution activities might lead to fine adjustment of the cluster toward its final structure, but did not affect cluster growth.

Taken together, our results suggest that numerous encounter events occurring at the cluster edge directly drive the steady growth of a social cluster, despite the highly dynamic and sporadic nature of these events.

## Multiple sensory modalities are required for cluster formation

To disentangle how sensory modalities mediate social aggregation, we quantified social clustering after selectively disrupting each sense, including vision, olfaction, gustation, audition and mechano-sensation. Previous studies suggested that visual input is essential for mediating aggregation in both larvae (*Dombrovski et al., 2017*; *Dombrovski et al., 2019*) and adult flies (*Simon et al., 2012*; *Burg et al., 2013*), suggesting that vision might be necessary in our paradigm. When depriving wild-type flies of visual inputs by testing under infrared light, SSI values markedly decreased, compared with those of control flies tested under regular white light (*Figure 5A*). Furthermore, flies with visual deficiency, *norpA*[33] (*Pak et al., 1970*) failed to form clusters, instead dispersing throughout the arena and exhibiting large social space, suggesting that the absence of visual cues impaired social aggregation (*Figure 5A*).

To test whether the olfactory pathway mediates social interaction during network formation, we analysed flies with impaired olfaction by mutating *Orco*, a gene that encodes an olfactory coreceptor in *Drosophila* (*Larsson et al., 2004*). The SIN analysis revealed that male *Orco* mutants moved slowly and exhibited a reduced ability to form SINs, but SINs that were formed had a higher proportion of reciprocated interactions and a longer average network distance between individuals (*Schneider et al., 2012*). On the other hand, the olfactory deficit in *Orco* males did not affect social space (*Simon et al., 2012*). Interestingly, while female *Orco* mutant flies exhibited lower SSI values, impaired olfaction in male *Orco* mutants had no effect on social clustering (*Figure 5B*). Furthermore, the impaired behaviour in *Orco*[-/-] females was rescued by re-expressing the *Orco* gene in the olfactory system (*Figure 5B*). To further study the involvement of olfactory inputs in social clustering, we performed surgical experiments to eliminate olfactory inputs, removing antennae and maxillary palps housing olfactory receptors (*Vosshall and Stocker, 2007*). As shown in *Figure 5B*, without antennae and maxillary palps, female flies, but not male flies, displayed dramatically decreased SSI, confirming the phenotypes in female *Orco* mutants. This result suggests the possibility that the olfactory modality plays a sexually dimorphic role in social clustering behaviour.

To assess whether the gustatory system is required for social interactions, we studied flies with mutations in the *Poxn* gene (*Dambly-Chaudière et al., 1992*). This mutation transforms sensilla that are typically destined for chemical detection into mechanosensory sensilla (*Awasaki and Kimura, 1997*; *Nottebohm et al., 1994*). Mutations in *Poxn* reduced the ability to form SINs, while the structural features of the SINs formed were similar to those without social interactions (*Schneider et al., 2012*). We found that *Poxn* mutant flies showed significantly impaired social cluster formation (*Figure 5C*). Although *Poxn* mutants exhibited strong deficits in forming networks of social

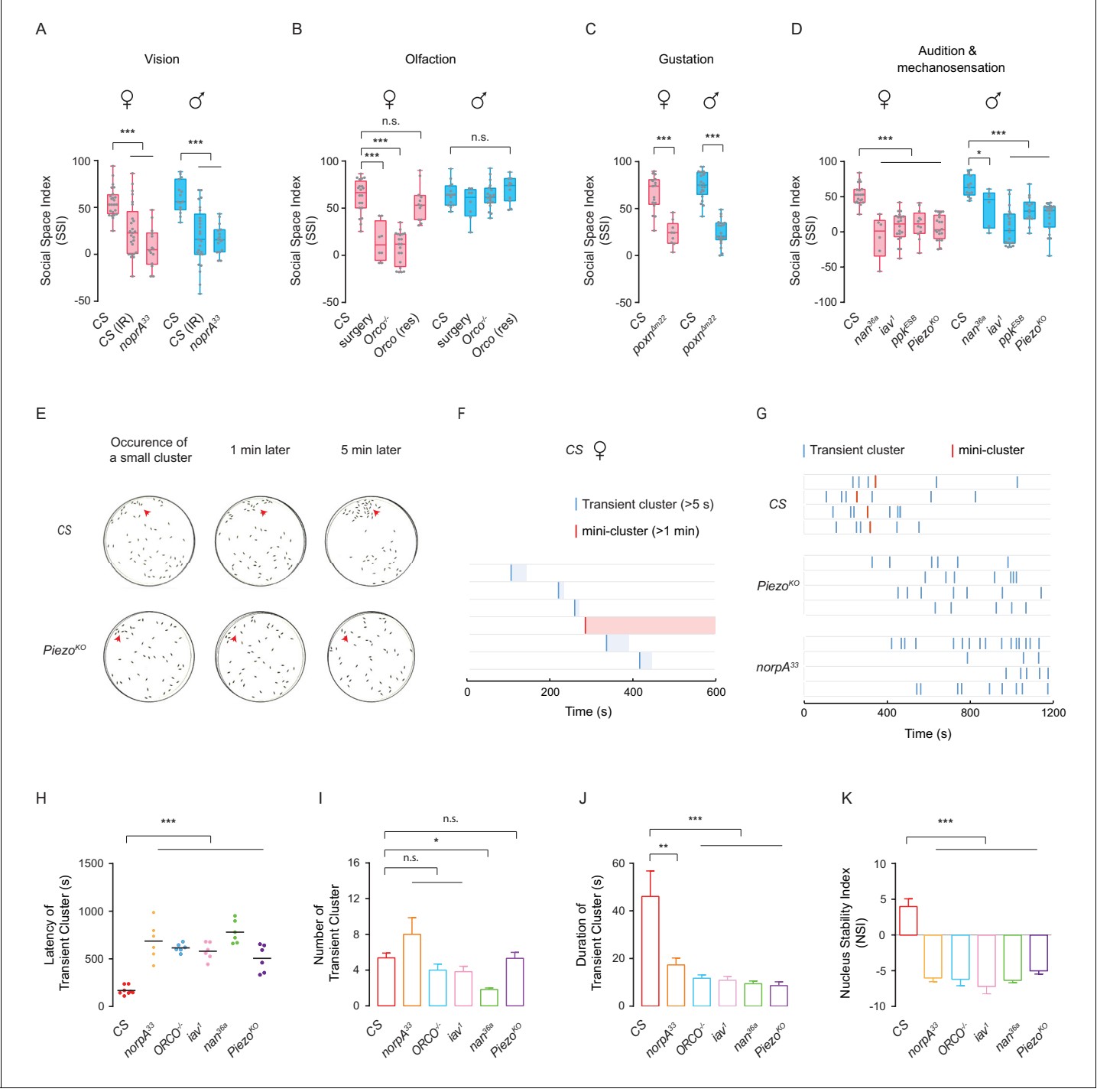

**Figure 5.** Sensory deficits impede formation of social clusters. (**A**) The levels of SSI in flies without vision: wild-type flies under dark (illuminated by arrays of infrared LED (850 nm) and the *norpA*[33] mutants (N = 10–25). (**B**) The levels of SSI of anosmic flies: wild-type flies without antennae and maxillary palps and the *Orco*[-/-] mutants (N = 8–20). (**C**) The levels of SSI in mutants with defective gustatory sensation (*Poxn*[Δm22]) (N = 9–26). (**D**) The levels of SSI in auditory/proprioceptive mutants (*inactive*[1] and *nanchung*[36a]) and nociceptive touch mutants (*ppk*[ESB] and *Piezo*[KO]) (N = 10–25). (**E**) Sequential images showing the spatial distribution of wild-type flies (top) and *Piezo*[KO] mutants (bottom) at 0, 1 and 5 min after the occurrence of a small cluster in each arena. Red arrows point to the sites of small clusters. (**F**) Representative data from an arena with the wild-type flies of indicating the onset and duration of transient clusters (blue) and mini-clusters (red). Transient clusters would be dissolved within 1 min, while a mini-cluster would last over 1 min, and served as the potential core to develop into a mature cluster. (**G**) Example data showing the events of transient clusters (blue) and mini-clusters (red) emergent in female *CS*, *Piezo*[KO] and *norpA*[33] groups during the first 20 min of observation. Four arenas for each genotype. (**H**) Latency of emergence of the first transient cluster in the flies of different genotypes during 20 min (N = 6–8 arenas). (**I**) Number of transient clusters in flies of

*Figure 5 continued on next page*

*Figure 5 continued*

different genotypes during 20 min (N = 6–8 arenas). (J) Average durations of transient clusters in different genotypes during 20 min (N = 6–8 arenas). (K) Comparison of Nucleus Stability Index in flies with indicated genotypes. NSI describes the change in the size of the nascent cluster within 1 min. All genotypes and experimental conditions are indicated with the plots. In a box and whiskers plot, scatter points show all data points, the whiskers mark minimum and maximum, and the middle line indicates the median of the data set. n.s. indicates not significant (p>0.05); ***: p<0.001, **: p<0.01 (Student's *t*-test within each genotype for two-group comparisons, one-way ANOVA with Dunnett's test for multiple comparisons to control [*CS*]). In a bar graph plot, error bars in (**H–K**) indicate s.e.m.

The online version of this article includes the following source data and figure supplement(s) for figure 5:

**Source data 1.** Figure 5A–D, H-K source data and related summary statistics.
**Figure supplement 1.** Evaluating social clustering with blockage of specific sensory inputs.
**Figure supplement 1—source data 1.** Figure 5—figure supplement 1A-B source data and related summary statistics.

---

interactions (*Schneider et al., 2012*) and failed to form clusters, we speculated that the drastic slow-walking of *Poxn* mutants would not lead to sufficient social encounter events for effective clustering. Therefore, instead of pursuing *Poxn* mutants further, we tested flies with disruptions of specific gustatory sensations, including chemical sensory related mutants $Gr33a^1$ (bitter) (*Moon et al., 2009*), $Gr64f^{-/-}$ (sweet) (*Jiao et al., 2008*), $Ir76b^1$ (salt and fatty acid) (*Ahn et al., 2017*; *Zhang et al., 2013a*) and contact pheromone related mutants $\Delta Gr32a^1$ (*Miyamoto and Amrein, 2008*) and $\Delta PPK23$ (*Lu et al., 2012*; *Toda et al., 2012*). All these mutant flies failed to form social clusters (*Figure 5—figure supplement 1A*). Taken together, these data suggest that multiple chemical cues are required for social clustering.

To investigate the contribution of mechanosensory inputs to social aggregation behaviour, we tested several mutants, including *inactive*, *nanchung* (*Gong et al., 2004*) (*Li et al., 2016*), *ppk* (*Adams et al., 1998*; *Zhong et al., 2010*; *Olds and Xu, 2014*) and *Piezo* (*Kim et al., 2012*) which participate in mechanosensation in *Drosophila*. As shown in *Figure 5D*, the flies with defective mechanosensation had a dramatically decreased level of cluster formation, compared with controls. It was suggested that *inactive* and *nanchung* functioned both for audition (in Johnston's organ) and for mechanosensation (in other chordotonal organs over the body) (*Gong et al., 2004*; *Karak et al., 2015*). Surgical removal of a pair of aristae to block auditory detection by Johnston's organ did not impair SSI (*Figure 5—figure supplement 1B*), excluding the mechanosensory contribution by Johnston's organ to social clustering.

Together, our surveys of basic sensory modalities suggested that social clustering depends on intact perception of multiple sensory cues, as disruption of any basic sensory input results in impaired cluster formation. The surprisingly high demand on precise perception of sensory cues was in accord with our observations that collective social behaviour was mediated by complicated interactions between highly mobile individuals within a constantly changing social environment.

## Abnormal encounter dynamic, rather than locomotion deficits, precluded cluster formation

To investigate why sensory disrupted flies failed to form social clusters, we examined the period of cluster initiation. We identified the occurrence of small clusters (aggregations of at least five flies in close proximity, within 1.5 body lengths), which would develop into social clusters by incorporating additional members. Small clusters arose in the group of mutant flies similar to wild-type flies, but were quickly disrupted, within minutes; for example the $Piezo^{KO}$ flies in *Figure 5E*. Accordingly, we classified small clusters into two types: transient clusters (stable for at least 5 s, but no more than 1 min) and mini-clusters (stable for over 1 min, which is the potential core of the cluster growth). In the wild-type population, while transient clusters occurred earlier and more frequently, a mini-cluster occurred once and lasted for the entire observation period (*Figure 5F*). Interestingly, in the population of $Piezo^{KO}$ and $norpA^{33}$, transient clusters appeared much later than wild-type, and a stable mini-cluster never emerged in mutant flies despite multiple occurrences of transient clusters (*Figure 5G*). It took substantially longer for occurrence of transient clusters in other mutants as well (*Figure 5H*). The success in forming social clusters by wild-type flies did not depend on the frequency of transient clusters (*Figure 5I*). However, compared with wild-type flies, these mutants showed shorter durations of transient clusters, suggesting an inability to maintain a budding cluster (*Figure 5J*). We introduced the Nucleus Stability Index (NSI) to quantify changes in the size of

nascent small clusters within 1 min. While nascent mini-clusters (nuclei) in wild-type flies were stable and grew steadily, small clusters in mutant flies shrank quickly and collapsed when flies dispersed (*Figure 5K*). These results demonstrated that sensory deficits affect the ability to develop or maintain a sizeable cluster.

To better understand the mechanisms underlying the ways in which sensory inputs mediate social clustering, we performed detailed analyses of encounter responses over the course of clustering in mutants including *norpA*, *Orco*, *inactive*, *nanchung*, and *Piezo*. Interestingly, comparing with wild-type flies, all of the mutant flies displayed a high frequency of social encounter events (*Figure 6A*) and a short duration of social interaction (*Figure 6B*), suggesting that the high encounter frequency in mutant populations did not lead to effective social clustering. Further quantification of behavioural responses after dyadic encounters revealed that mutant 'interactors' exhibited a similar tendency to 'move away' after encounters compared with wild-type flies (*Figure 6C*). Conversely, mutant 'interactees' exhibited a significantly reduced tendency to 'stay' (*Figure 6C*). Considering the net results for encounters in mutants, the difference between the likelihood for both flies to stay and for both to move away, it was more frequent for both flies to 'move' than for both to 'stay' (*Figure 6D*). Therefore, the mutant pairs dispersed more effectively away from the encounter site.

To understand the encountering process in more detail, we analysed the transient velocities of flies before and after encounter events (+ /- 0.8 s). 'Interactors' exhibited decreased velocity before impact (*Figure 6E–F*) while 'interactees' exhibited increased velocities after encounters (*Figure 6G–H*). Compared with wild-type flies, mutant flies generally exhibited higher transient speeds either before or after encounters (*Figure 6E–H*).

To confirm that these mutant flies are capable of walking faster than wild-type flies, we observed the spontaneous walking and exploration of individual flies in a dish (diameter: 90 mm). As shown in *Figure 6—figure supplement 1A–C*, large proportions of mutant flies exhibited normal or faster spontaneous locomotive speeds (walking with higher average and maximum speeds). This strongly suggested that the inability to achieve formation of social clusters in these mutants was not due to locomotion deficits. In addition, the overall profiles of change of speed (acceleration and deceleration) in mutants were similar to or higher than the wild-type flies (*Figure 6—figure supplement 1D–E*). Importantly, compared with wild-type flies, the rapid change of locomotion speed (reflected through average- and maximum- acceleration [and deceleration]) in these mutants strongly suggested that sensory disruptions did not affect their locomotion controls when walking on a horizontal surface (*Figure 6—figure supplement 1F–G*).

Therefore, abnormal social interactions stemmed from defective inter-fly communications, rather than locomotion per se, contributing to unsuccessful formation, maintenance or growth of social clusters. The increased population dynamics, including high locomotion speed, frequent encounters, short interaction duration, and effective dispersion after encounter, in mutants likely indicated their elevated but failed attempts to compensate for the loss of social cues mediating cluster formation.

## *ppk*-specific neurons participate in establishing normal social space

Stereotypic responses from dyadic interactions helped to build up the social cluster, these local interaction events highlighted the importance of physical contact in contributing to the clustering process, possibly by transducing mechanical and chemical signals. To further examine the neuronal basis of social clustering, we used neurogenetic approaches to modulate activities in target neurons while analysing behaviour change. The ubiquity of appendage-touch in eliciting encounter response prompted us to screen neurons potentially occurring in appendage organs using specific GAL4s, including all olfactory receptor-GAL4s, gustatory receptor-GAL4s, GAL4s labelling ion channels participating in mechanosensation in *Drosophila* including *ppk* (*Adams et al., 1998*; *Zhong et al., 2010*; *Olds and Xu, 2014*), *inactive*, *nanchung* (*Gong et al., 2004*; *Karak et al., 2015*; *Li et al., 2016*), *nompC* (*Walker et al., 2000*; *Yan et al., 2013*) and *Piezo* (*Kim et al., 2012*), and GAL4s targeted to the systems of neurotransmitters, neuropeptides, mechanosensation and those based on the expression patterns of a GAL4 driver line resource (*Jenett et al., 2012*; *Ramdya et al., 2015*; *Tuthill and Wilson, 2016*; *Mamiya et al., 2018*).

We utilised the GAL4/UAS binary system to express an optogenetic activator, CsChrimson (*Klapoetke et al., 2014*), to forcibly activate selected neurons during social clustering. Interestingly, among the approximately 500 lines screened, *ppk >CsChrimson* flies exhibited extensive aggregation with significantly decreased social space in male and female flies (*Figure 7A,B,D*; *Figure 7—*

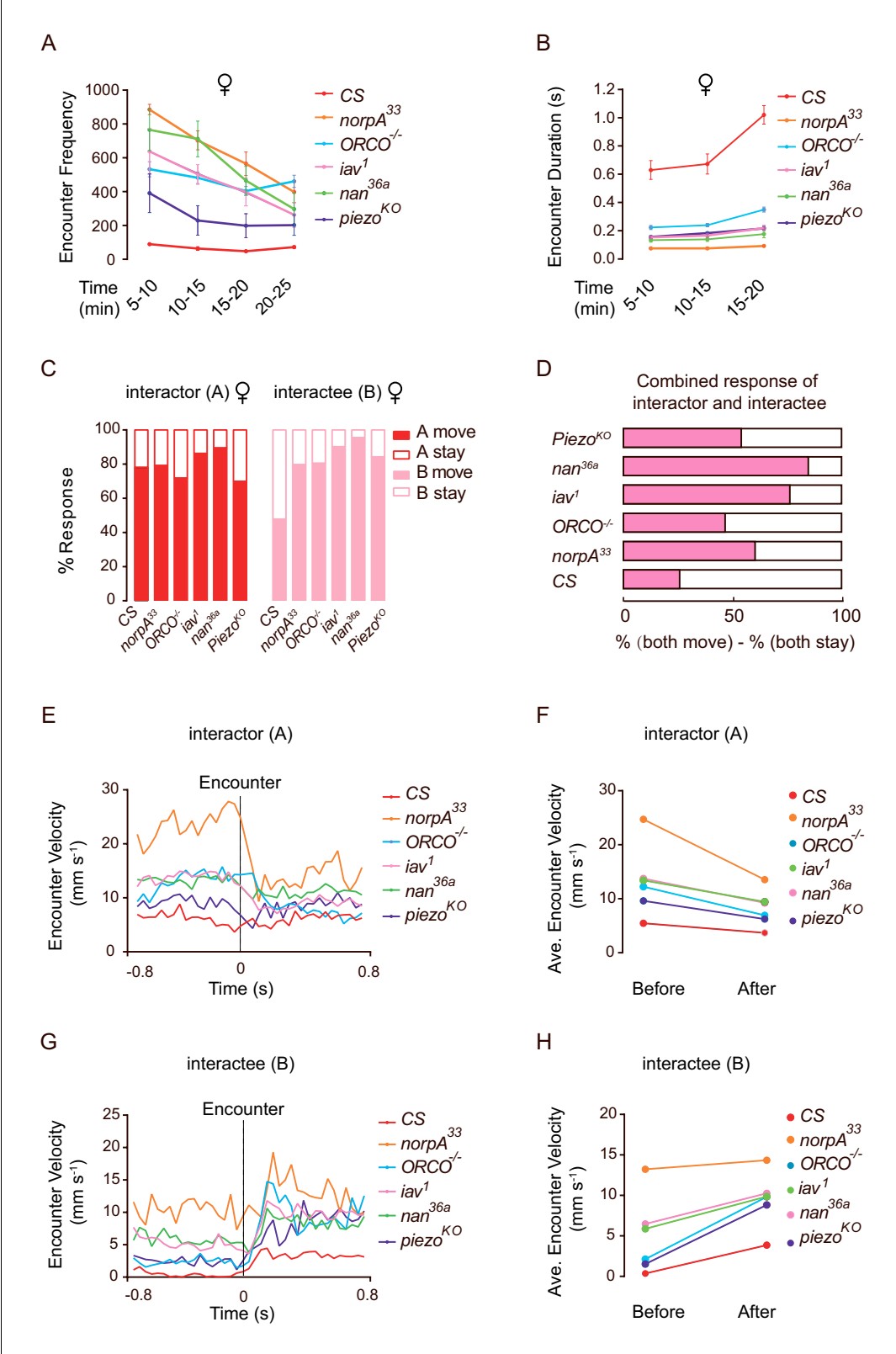

**Figure 6.** Mutant flies exhibit abnormal encounter dynamics. (**A**) Quantification of the number of encounter events during the indicated time periods in female flies of different genotypes (N = 4 arenas, n = 1090–10,205 encounters). (**B**) Average duration of encounters in different genotypes (60–120 encounters). The duration of an encounter is the time length from the beginning of physical contacts of two flies to the end of their last contact. (**C**) Percentage of behavioural outputs after encounter of the Interactor (A, red) and Interactee (B, pink), 'A move' and 'B move' indicate flies showing

*Figure 6 continued on next page*

*Figure 6 continued*

changes in locations after encounter, while 'A stay' and 'B stay' indicate flies that stayed at the encountering location (N = 3–5 arenas, n = 60–100 encounters). (D) Net behavioural output of the encounter events in (C). If only one fly moved away (either 'A stay and B move' or 'A move and B stay') would not change the number of flies at the encounter site since before the encounter, one fly ('A') walked into the site. Thus, we compared the likelihood of both A and B staying with that of both A and B moving away. The bar graph shows the difference in the percentages of these two types of outputs for each genotype (N = 3–5 arenas, n = 60–100 encounters). (E) The transient velocities of Interactors of different genotypes during the course of encounter. (N = 3 arenas, n = 30–60 encounters). (F) The average velocities of Interactors before and after encounter (N = 3 arenas, n = 30–60 encounters). (G, H) The transient velocities (G) and average velocities (H) of Interactees before and after encounter (N = 3 arenas, n = 30–60 encounters). All genotypes and experimental conditions are indicated with the plots. Error bars in (A–B) indicate s.e.m.

The online version of this article includes the following source data and figure supplement(s) for figure 6:

**Source data 1.** Figure 6A–C, E-H source data and related summary statistics.
**Figure supplement 1.** Mutants failed to form social clusters, but were capable of locomotion manoeuvers.
**Figure supplement 1—source data 1.** Figure 6—figure supplement 1B-C, F-G source data and related summary statistics.

*figure supplement 1A,B*). Optogenetic activation of the mechanosensory neurons in *iav >CsChrimson*, *nompC > CsChrimson* or *Piezo > CsChrimson* flies reduced SSIs, while *nan >CsChrimson* flies behaved normally as wild-type controls (*Figure 7—figure supplement 1A,B*).

Moreover, to block the functions of mechanosensory neurons, we expressed tetanus toxin light chain (TNT) (*Sweeney et al., 1995*) in different candidate neurons. All flies with the corresponding neurons being silenced failed to form social clusters (*Figure 7C,E*; *Figure 7—figure supplement 1C,D*), suggesting that the activities of these neurons, including *ppk-GAL4*-labelled neurons, are required for cluster formation. Notably, video sequences of cluster development revealed that activating *ppk*-labelled neurons resulted in early formation of clusters, indicating that the emergence of social clusters was faster in *ppk >CsChrimson* flies than in wild-type flies (*Figure 7F*). However, the locomotion activity of *ppk >CsChrimson* flies was comparable to that of genetic controls (*Figure 7G*), suggesting that faster clustering is unlikely to be due to faster walking.

The reporter protein green fluorescent protein (GFP) revealed that ppk-specific GAL4 labelled various populations in the peripheral nervous system, including legs and wing margins (*Figure 7H–K*), whereas *iav-GAL4*, *nan-GAL4*, *nompC-GAL4* and *Piezo-GAL4* labelled limited numbers of neurons scattered in the appendages (*Figure 7—figure supplement 2*). These diverse expression patterns suggest potentially complementary roles of these mechanosensory neurons in social encounters via appendage-touches to mediate social communication.

Together, our results indicated that *ppk*-specific neurons play a unique role in regulating cluster formation and social distance.

## Social grouping elevates activity in tarsal *ppk* neurons

Two findings, that appendage-touch played an important role in social encounters and that *ppk* was broadly expressed in the appendage organs, led us to investigate *ppk* neurons in appendages in more detail. We utilised an activity reporter system, calcium-dependent nuclear import of LexA (CaLexA) (*Masuyama et al., 2012*), to assess whether the activity of *ppk* neurons correlates with inter-fly physical interactions. Female flies were reared under isolated or social conditions, and the activities of *ppk* neurons were subsequently quantified. As shown in *Figure 8A–C*, flies raised in a group exhibited significantly increased activities in a small group of *ppk* neurons in the tip of the tarsus, compared with those raised alone. In contrast, there were no differences in CaLexA signals in neurons on the wing margin under single- or group-raised conditions (*Figure 8—figure supplement 1A–D*). Notably, grouping previously singly-raised flies together for 30 hr evoked substantial activity in *ppk* neurons in the tarsus (*Figure 8B,C*), suggesting that frequent inter-fly interactions involving appendage-touches while living in a group might increase the activity of *ppk* neurons in the tarsus.

To discern the possible effects of contact pheromones, volatile chemicals and physical touch under grouped conditions, we raised a fly in a group but deprived it of direct contact with others. As shown in *Figure 8D*, each 'netted in a group' fly was grown in a small case and separated from the other flies via a double-layered net on top of the cage, while, as a control, 'netted single' flies were not in the presence of other flies outside the net. The net allowed odours, sounds and certain visual information to pass through, but blocked contact-dependent cues. Surprisingly, both 'netted single' and 'netted in a group' flies showed similar CaLexA signals in the tarsal *ppk* neurons as

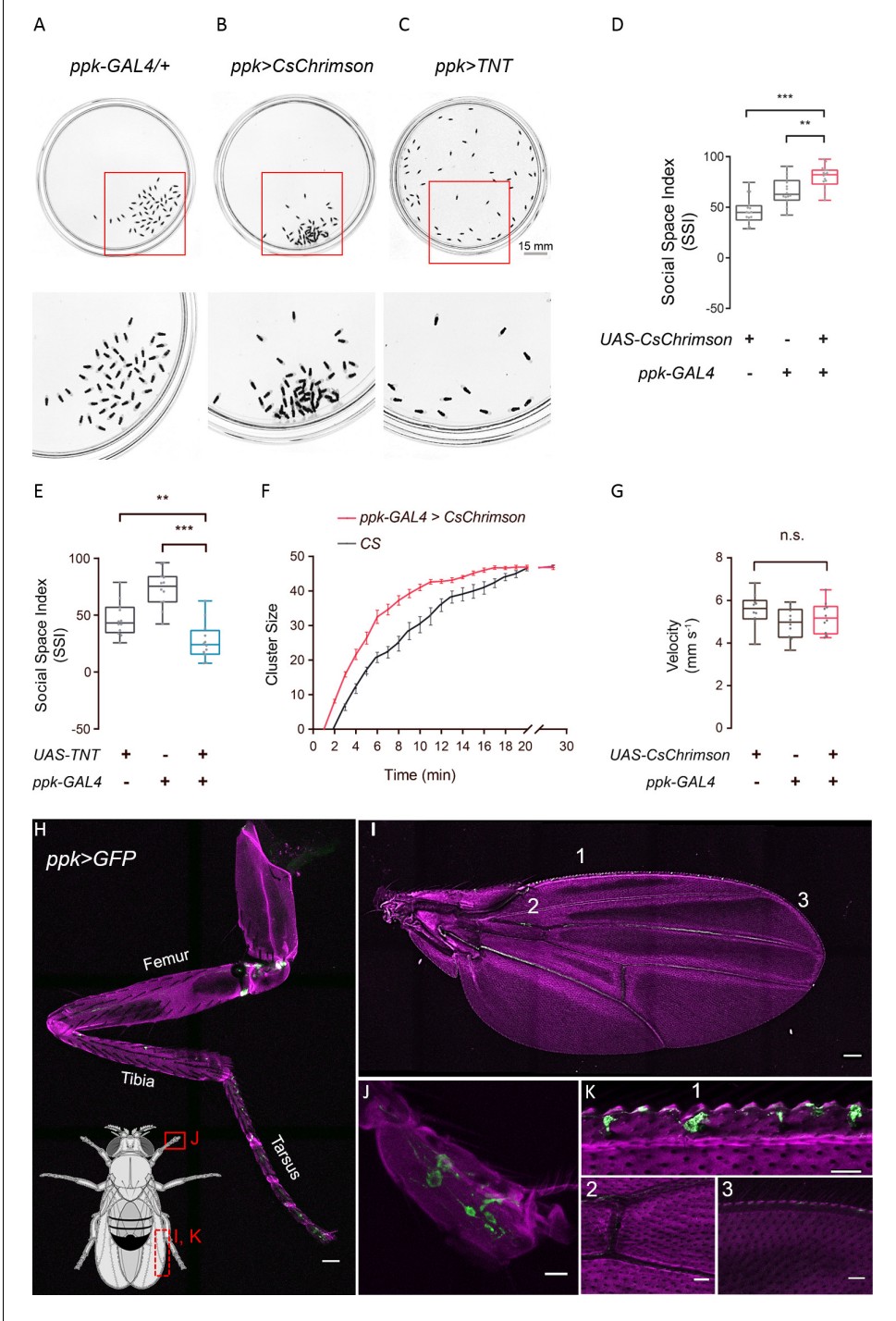

**Figure 7.** *ppk*-specific neurons are important for social cluster and social space. (**A–C**) Representative images showing the spatial distributions of flies of genetic control (**A**), flies with optogenetically-activated ppk neurons (**B**), and flies with silenced ppk neurons (**C**). Bottom: the enlarged views of regions in corresponding arenas on the top, marked by red squares. (**D**) SSIs of flies with optogenetic activation of *ppk* neurons (red) and genetic controls (grey). (N = 16 arenas). (**E**) SSIs of flies with silenced *ppk* neurons (blue) and genetic controls (grey) (N = 16 arenas). (**F**) Comparing the cluster sizes (number of flies in a cluster) of female flies over the course of clustering (N = 8 arenas). (**G**) Comparing the locomotion speed in flies with optogenetically activated *ppk* neurons (red) and genetic controls (grey) (N = 3 arenas, n = 30 flies). (**H–K**) Expression pattern of ppk-GAL4 in the peripheral. Green channel shows GFP signals from *ppk-GAL4 >UAS-mCD8-GFP* and magenta channels show autofluorescence from the cuticle. Body parts shown are: foreleg (H, scale bar: 50 μm), wing (I, scale bar: 100 μm), tip of the tarsus (J, scale

*Figure 7 continued on next page*

*Figure 7 continued*

bar: 10 μm), and portions of a wing (K1: pre-wing margin, K2: vein in wing, K3: post-wing margin, scale bars in K2-3: 30 μm). All genotypes and experimental conditions are indicated with the plots. In a box and whisker plot, scatter points show all data points, the box includes the 25th to 75th percentiles, the whiskers show the minimum and maximum, and the middle line indicates the median of the data set. n.s. indicates not significant (p>0.05); **p<0.01, ***: p<0.001 (Student's *t*-test). Error bars in (**F**) indicate s.e.m.

The online version of this article includes the following source data and figure supplement(s) for figure 7:

**Source data 1.** Figure 7D–G source data and related summary statistics.
**Figure supplement 1.** Quantification of social clustering after activating or silencing neurons related to mechanosensory function.
**Figure supplement 1—source data 1.** Figure 7—figure supplement 1A-D source data and related summary statistics.
**Figure supplement 2.** Expression patterns of mechanosensory GAL4s in the wings and legs.

---

singly-raised flies (*Figure 8D,E*), while re-grouping the 'netted single' flies for 24 hr was sufficient to induce an increase in CaLexA signals (*Figure 8D,E*).

Together, these data suggested that the *ppk*-labelled neurons in the tarsus responded to physical touch, specifically from other flies. The elevated activities of tarsal *ppk* neurons from social contact highlighted their function in dyadic encounters to mediate appendage touches, which are important for social clustering.

## Discussion

In this study, we established a simple collective behaviour paradigm using *Drosophila* as a model system to investigate how loosely distributed individuals come together to form a stable and orderly social cluster. Thus, we linked a dynamic progress (collective physical interactions) with its outcome (a structured cluster). We demonstrated that a group of individuals self-organised into a social cluster with topological features of a distributed network. This process relied on multiple sensory modalities as well as internal state and previous social experience. Moreover, the order emerged dynamically from stereotypical responses of numerous asymmetric encounters between pairs of individuals, while, abnormal encounter responses exhibited by mutant flies resulted in a failure of social clustering. These dyadic interactions, particularly near the border of a cluster, contributed to cluster growth, eventually incorporating all individuals. Physical contact in a group induced dramatic increases in the activity of tarsus *ppk* neurons, the activation of which caused flies to group quickly and form more compact clusters, demonstrating the important roles of *ppk* neurons in mediating dyadic interactions to promote clustering.

Many species exhibit massive collective behaviour, such as starling flocks (*Nagy et al., 2010*), schools of blue jack mackerel (*Bertrand et al., 2004*), stickleback shoals (*Jolles et al., 2017*), swarms of desert locusts (*Buhl et al., 2006*), and swarms of *C. elegans* (*de Bono and Bargmann, 1998*; *de Bono, 2003*). While *Drosophila* are commonly considered 'solitary' (*Guo et al., 2017*), increasing evidence suggests that *Drosophila* form social groups (*Stalker and Harrison, 1961*; *Navarro and del Solar, 1975*; *Simon et al., 2012*; *Schneider et al., 2012*; *Burg et al., 2013*; *Ramdya et al., 2017*; *Guo et al., 2017*). The lack of strikingly visible collective behaviour like that exhibited by birds and fish is made up for by the unique advantages of *Drosophila* for experimentally investigating the neural mechanisms underlying collective actions, including simple brain structure, tractable behaviour and a smaller genome. Additionally, in our paradigm, flies interacted collectively in a defined two-dimensional space, whereas many collective behaviours, including those of starlings and locusts, occur in an open three-dimensional space; thereby, our system reduced the difficulties of modelling. Furthermore, in a laboratory setting, we took into account the location and locomotion of all flies from the beginning, when individuals were still dispersed, to the end of the process, when all individuals joined to form a social cluster. Moreover, working with fruit flies allowed us to control many influential factors, including temperature, time of day, diet, physiological status of the animals (such as genetic background, age, and gender) and previous social experience. It is extremely difficult to control these factors in wild animals, despite the importance of such controls for social behaviour analysis. Homogenous data can help identify principal components by reducing undesirable variation

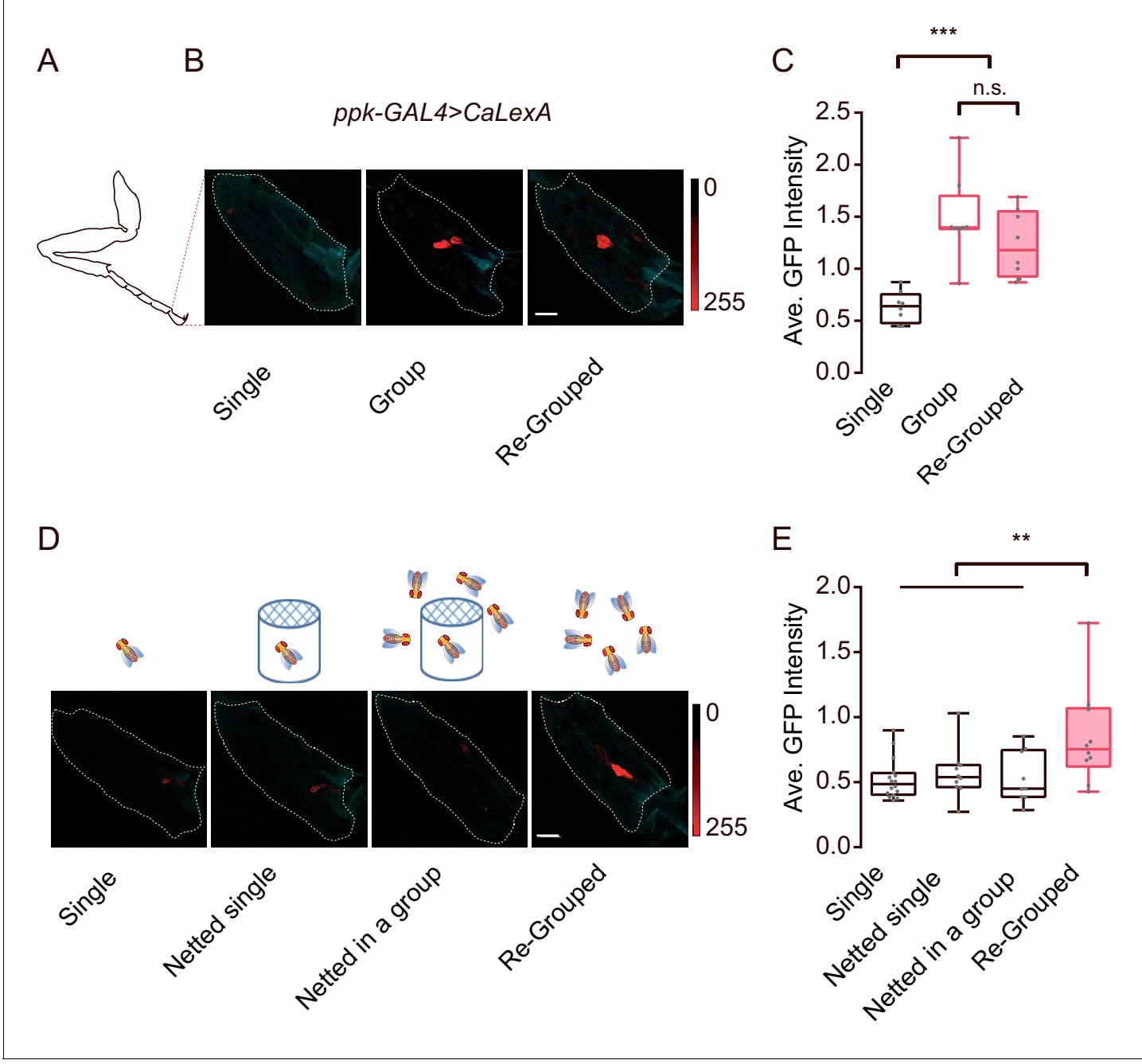

**Figure 8.** Contact-dependent activation of *ppk*-labelled neurons by social grouping. (**A**) A schematic showing the imaging area (tip of tarsus) for (**B**) and (**D**). (**B**) Representative images showing CaLexA signals in *ppk* neurons of female flies with different social experience. Single: individual flies were raised in isolation for 16 days after eclosion; Group: flies were raised in a group for 16 days after eclosion; Re-grouped: 10 singly-raised flies were combined together and maintained for 30 hr. Cyan: autofluorescence from the cuticle, Red: maximal intensity of CaLexA signals. Dashed lines trace the tip of tarsus as ROIs for calculating GFP signals. Scale bar, 10 μm. (**C**) Comparing the average intensity of CaLexA signals in *ppk* neurons from flies treated with indicated conditions shown in (**B**) (N = 8 arenas). (**D**) Schematics of different treatments (top) and the corresponding representative images (bottom) of the tip of the tarsus of *ppk-GAl4 >CaLexA* flies. Single: flies raised in single isolation for 16 days after eclosion; Netted single: single fly raised in a netted tube (diameter: 12 mm, covered by double-layered net on the top end) alone for 16 d; Netted in a group: single fly raised in a netted tube which was surrounded 50 flies (inaccessible) for 16 d; Re-grouped: 10 singly-raised flies were combined and maintained for 24 hr. Cyan: autofluorescence from the cuticle, Red: maximal intensity of CaLexA signals. Dashed lines trace the tip of the tarsus (as a ROI for computing the GFP signals). Scale bar, 10 μm. (**E**) Comparing the average intensity of CaLexA signals in *ppk* neurons from flies treated with indicated conditions shown in (**D**) (N = 8 arenas). All genotypes and experimental conditions are indicated with the plots. In a box and whisker plot, scatter points show all data

*Figure 8 continued on next page*

*Figure 8 continued*

points, whiskers mark minimum and maximum, and the middle line indicates median of the data set. n.s. indicates not significant (p>0.05); **: p<0.01, ***: p<0.001 (Student's *t*-test within each genotype for two-group comparisons, one-way ANOVA with Tukey's post hoc test for multiple comparisons). The online version of this article includes the following source data and figure supplement(s) for figure 8:

**Source data 1.** Figure 8C, E source data and related summary statistics.
**Figure supplement 1.** The activity of *ppk*-labelled neurons of wing margins under different treatments.
**Figure supplement 1—source data 1.** Figure 8—figure supplemnet 1B source data and related summary statistics.

and noise. *Drosophila* offers a simpler alternative for examining the basic principles of collective behaviour by modelling with data from field observations.

Although neural substrates in the peripheral and central nervous systems for controlling interactions between a pair of flies (such as in typical courtship and aggression behaviour) are increasingly well understood, little is known about the dynamics and structures of social groups in fruit flies. Built upon the investigations of aggregation of a group of *Drosophila* (*Simon et al., 2012*; *Burg et al., 2013*), our improved paradigm generates a relatively naturalistic environment with minimal perturbations to the flies, and reveals the unique structure of fly assemblage. The robust tendency to form organised social clusters provides an interesting entry point to investigate collective behaviours in fruit flies. As rudimentary as a *Drosophila* cluster appears compared with a swarm-like flock of starlings, the flies were not simply clumped together in a random manner. Rather, the arrangement of flies in clusters strongly resembled that of a distributed network, rather than a centralised or decentralised network (*Baran, 1964*). Previous studies revealed that maintaining appropriate social space in flies requires visual input (*Simon et al., 2012*; *Burg et al., 2013*), and is modulated by previous social experience (*Simon et al., 2012*) and anaesthetic treatment (*Burg et al., 2013*). Our quantitative behavioural analysis, in conjunction with neurogenetic approaches and surgical experiments, revealed that cluster formation by groups of flies is a highly dynamic process, requiring integration of visual, olfactory, gustatory, auditory and mechanosensory inputs. Impaired sensory functions resulted in abnormal encounter responses, and, in turn, a failure in cluster formation. Notably, we found sexual dimorphism in olfactory-mediated social aggregation, suggesting that, in male and female flies, different neural circuits regulate the process of social aggregation. Importantly, our data suggested that different sensory pathways are not functionally redundant, and even different mechanosensory neurons play different roles in shaping the formation and spacing of social clusters. Taken together, our results indicate that social clustering behaviour in fruit flies is more sophisticated than previously assumed and a model of dynamic integration of various sensory inputs is required to explain group-level aggregation behaviour.

We modelled how a fly behaves at different distances in response to influences from others, using a force field computed from putative attractive and repulsive forces. The attraction, likely mediated by vision and olfaction, is generally weak over a broad range of distances, whereas repulsion, likely through mechanosensation and gustation, is strong at short distances. The simulation results suggested that the antagonistic interaction between these forces is sufficient to generate the required social distance for cluster formation. Notably, Turing first proposed a theoretical model utilizing local activation and a long-range inhibition to explain a variety of patterns in biology (*Turing, 1952*). Similar models were developed to account for the self-organing pattern formation from cells to organisms, suggesting that diverse patterns of social structures do not necessitate the assumption of complex behavior (*Detrain and Deneubourg, 2006*). However, without the perspective of dynamic processes, our static models cannot account for the active process of pair-wise encounters. Our results demonstrated that a large number of random dyadic interactions can serve as a driving force for the emergence of an orderly social cluster. Generalized into a system of self-propelled particles (SPP), flies are likely to achieve clustering by a parallel process involving all individuals, each of which has only a strong local view, without a clear sense of other individuals or the final structure of the cluster.

Although the timing and locations of pair-wise encounters are highly variable, the types and consequences of these encounters are stereotypic, and, importantly, the net outcome of these collective events over time is the growth of a cluster to include all flies in the arena. Interestingly, in wild-type flies, the frequency of encounters decreases after the initiation of the social cluster, indicating that

an increasing number of flies settle down through dyadic interactions. In addition, a large proportion of flies gradually joined the nascent cluster (nucleus), despite its sporadic occurrence and erratic location. Similar approaches have been observed in 'food searching-aggregation' behaviour of adult flies, in which signals were emitted from 'primers' for the subsequent arrival of 'followers' (*Tinette et al., 2004*), and in cooperative digging behaviour of larvae (*Ben Jacob et al., 2004*; *Dombrovski et al., 2017*; *Dombrovski et al., 2019*). Besides foraging, grouping together benefits individuals in other aspects of *Drosophila* life. For example, evidence suggests that grouping in flies can synchronise circadian rhythms (*Levine et al., 2002*), modulate gene expression (*Krupp et al., 2008*), change pheromone profiles (*Krupp et al., 2008*; *Liu et al., 2011*), facilitate social learning (*Chabaud et al., 2009*; *Battesti et al., 2012*) and generate a heightened awareness of environmental stressors (*Ramdya et al., 2015*). Therefore, our data support the existence of a generalised assembling strategy for a group of flies to form a well-structured social network with or without external stimuli.

Although flies also exhibit pairwise interactions during courtship and aggression, the dyadic actions of encounter events during social clustering are relatively distinct in several aspects. First, the duration of each encounter event is brief, lasting for less than 1 s. Second, two flies from an encounter event are unlikely to meet again within a short period. This might be partially due to the sheer number of flies in the arena. However, in settings with many flies, individuals during their aggression or courtship exhibit a strong tendency for object fixation, as the pairs chase and interact repeatedly despite surrounding flies. Third, the frequent mode of dyadic encounter is between a walking fly and a stationary fly. The typical output of encounter is one moving-away while the other staying. During courtship and aggression, however, it is more common for both flies to be moving before and after the interaction. The distinct behavioural characteristics of encounter events suggest that, during social clustering, flies are likely exhibiting a unique 'mental' state.

Social clustering is not the only instance in which a fly group uses pairwise interactions to convey information between individuals. Benton and colleagues demonstrated that enhanced $CO_2$-avoidance of a group of flies over individual performance also depends on pairwise interactions (*Ramdya et al., 2015*). It is likely that additional situations exist in which *Drosophila* exhibit stereotypic pairwise interactions that serve as general driving force for the emergence of order from an initially disordered group. In ants, spontaneously mobilized individuals in nests can contact and excite other ants thereby spreading rhythmic activity across the colony (*Couzin, 2009*). In swarms of insects, schools of fish and flocks of birds, once the entire group accomplishes pattern formation and moves cohesively with high dynamics, an individual gathers information from near neighbors to adjust its own speed and direction (*Sumpter et al., 2008*; *Buhl et al., 2011*; *Katz et al., 2011*). In these examples to maintain a relatively uniformed distance between moving individuals, spreading information depends on non-contact pairwise interactions, unlike those seen in *Drosophila* mediated via physical touches. Previous studies have suggested that an individual engaged in collective movements 'averages' its pairwise interactions with near neighbors, although features of residual three-body interactions are also present (*Katz et al., 2011*; *Herbert-Read, 2016*). Furthermore, it is possible that pairwise interactions play roles before collective movements unfold, such as to accumulate a sufficient number of individuals into a small space to escalate the group density beyond a threshold. or to influence the collective mood or decision-making of the group for departure (*Petit and Bon, 2010*). Overall, there appear to be common rules in the collective behaviors of flies and other animals. Information is processed and exchanged at a local scale between near neighbors, and individuals do not (and do not need to) have a global view (*Buhl et al., 2011*). Furthermore, ordered organization only emerges when the group reaches to a certain size (*Buhl et al., 2006*).

Inspection of all dyadic encounters in our analysis demonstrated that touches of peripheral appendages between a pair of flies evoke stay-or-go responses, which is an important element during the initiation, growth and maintenance of the SIN. Similarly, cascades of appendage-touch elicited interactions between fly pairs are reported to be responsible for the collective avoidance of $CO_2$ (*Ramdya et al., 2015*). *Drosophila* possess an extensive mechanosensation system involving different types of mechanosensory neurons (*Karkali and Martin-Blanco, 2017*). Remarkably, our results demonstrated that manipulating neural activities of mechanosensory neurons caused aberrant social clustering, suggesting an important role of mechanosensation in regulating social networks in adult fruit flies. Considering the broad expression of mechanosensory genes, it is possible that multiple mechanosensory pathways mediate inter-fly tactile interactions in a coordinated and/or redundant

way. Ants are also reported to use antennae to communicate with each other via tactile signals and chemosensation (*Billen, 2006*). In addition, crowding of solitary locusts induces transformation into the gregarious phase. Tactile signals via appendage touches between individuals play an important role in this process (*Höltje and Hustert, 2003*; *Buhl et al., 2006*; *Bazazi et al., 2008*). Interestingly, we found that in grouped flies, the activity of *ppk* neurons was elevated via a contact-dependent mechanism, suggesting a conserved role of tactile perception in social grouping. Taken together, our study not only revealed a novel role of appendage-mediated tactile signals in the dynamic formation of social networks in flies, but also provides evidence of common features of collective behaviours among different species.

Our modelling and observations indicated that animals use attractive signals to assemble a large group, whereas repulsive signals between individuals prevent them from forming a jumbled mass. Although this study did not pinpoint the attractive signal(s), the regular pattern of social clusters and the ubiquity of appendage interactions between pairs of flies strongly suggest that a stereotypic contact-mediated repulsive force keeps the nearby flies apart at a 'hard-core' distance. Mechanosensation is likely to be necessary for flies to both exert and receive physical forces during a dyadic interaction. The tarsal *ppk* neurons are activated during direct social contact, suggesting the possibility that they may sense social tactile cues. The expression patterns of *ppk*-positive neurons in the appendages suggest the possibility that the normal function of these neurons is to perceive touches from other flies. However, it is also possible that *ppk*-neurons are also required on the force-exerting side, for example, to gauge or control the level of force generated to repel other flies. Activating *ppk* neurons leads to decreased repulsion between flies supports the second hypothesis. Detailed dissection of the roles of each type of *ppk*-neuron necessitates manipulating the subclasses with high spatial and temporal precision, which should be possible with future genetic tools (*Jenett et al., 2012*).

In conclusion, we developed a self-organised aggregation paradigm as a model for studying the emergence of ordered structures from collective interactions of individuals. Our findings highlight the importance of stereotypic dyadic interactions and *ppk* neurons in social clustering and spacing. Further, comprehensive exploration of the dynamics and guiding principles in social clustering in *Drosophila* could enable the development of a novel framework for understanding the molecular and neural bases of collective behaviour in other animals.

# Materials and methods

## Key resources table

| Reagent type (species) or resource | Designation | Source or reference | Identifiers | Additional information |
|---|---|---|---|---|
| Strain (*Drosophila melanogaster*) | *Canton-S* | (*Zhan et al., 2016*) | | https://doi.org/10.1038/ncomms13633 |
| Strain (*Drosophila melanogaster*) | *w*$^{1118}$ | Bloomington Drosophila Stock Center | RRID: BDSC5905 | |
| Strain (*Drosophila melanogaster*) | *ppk-GAL4* | Bloomington Drosophila Stock Center | RRID: BDSC32079 | |
| Strain (*Drosophila melanogaster*) | *UAS-CsChrimson* | Bloomington Drosophila Stock Center | RRID: BDSC55135 | |
| Strain (*Drosophila melanogaster*) | *Piezo-GAL4* | Bloomington Drosophila Stock Center | RRID: BDSC58771 | |
| Strain (*Drosophila melanogaster*) | *ORCO-GAL4* | Yi Rao lab, Peking University | | |

*Continued on next page*

*Continued*

| Reagent type (species) or resource | Designation | Source or reference | Identifiers | Additional information |
|---|---|---|---|---|
| Strain (*Drosophila melanogaster*) | *nan-GAL4* | Yi Rao lab, Peking University | | |
| Strain (*Drosophila melanogaster*) | *iav-GAL4* | Yi Rao lab, Peking University | | |
| Strain (*Drosophila melanogaster*) | *nompC-GAL4* | Yi Rao lab, Peking University | | |
| Strain (*Drosophila melanogaster*) | *UAS-TNTE* | Aike Guo and Yan Li lab (*Liu et al., 2016*) | | http://dx.doi.org/10.7554/eLife.13238.001 |
| Strain (*Drosophila melanogaster*) | *Tub-GAL80$^{ts}$* | Aike Guo and Yan Li lab (*Liu et al., 2016*) | | http://dx.doi.org/10.7554/eLife.13238.001 |
| Strain (*Drosophila melanogaster*) | *Cha3.3kb-GAL80* | Aike Guo and Yan Li lab (*Zhang et al., 2013b*) | | https://doi.org/10.1523/JNEUROSCI.5365-12.2013 |
| Strain (*Drosophila melanogaster*) | CaLexA | Jing Wang lab (*Masuyama et al., 2012*) | | https://dx.doi.org/10.3109%2F01677063.2011.642910 |
| Strain (*Drosophila melanogaster*) | UAS-mCD8::GFP | (*Zhan et al., 2016*) | | https://doi.org/10.1038/ncomms13633 |
| Gene (*Drosophila melanogaster*) | *norpA$^{33}$* | Bloomington Drosophila Stock Center | RRID: BDSC9047 | |
| Gene (*Drosophila melanogaster*) | *Gr64f$^{-/-}$* | Bloomington Drosophila Stock Center | RRID: BDSC27883 | |
| Gene (*Drosophila melanogaster*) | *Δppk23* | Bloomington Drosophila Stock Center | RRID: BDSC33300 | |
| Gene (*Drosophila melanogaster*) | *Gr33a$^{1}$* | Bloomington Drosophila Stock Center | RRID: BDSC31427 | |
| Gene (*Drosophila melanogaster*) | *IR76b$^{1}$* | Bloomington Drosophila Stock Center | RRID: BDSC51309 | |
| Gene (*Drosophila melanogaster*) | *Piezo$^{KO}$* | Bloomington Drosophila Stock Center | RRID: BDSC58770 | |
| Gene (*Drosophila melanogaster*) | *nan$^{36a}$* | Bloomington Drosophila Stock Center | RRID: BDSC24902 | |
| Gene (*Drosophila melanogaster*) | *ORCO$^{-/-}$* | Yi Rao lab, Peking University | | |
| Gene (*Drosophila melanogaster*) | *UAS-ORCO* | Yi Rao lab, Peking University | | |
| Gene (*Drosophila melanogaster*) | *iav$^{1}$* | Yi Rao lab, Peking University | | |
| Gene (*Drosophila melanogaster*) | *Poxn$^{Δm22}$* | Yi Rao lab, Peking University | | |

*Continued on next page*

*Continued*

| Reagent type (species) or resource | Designation | Source or reference | Identifiers | Additional information |
|---|---|---|---|---|
| Gene (*Drosophila melanogaster*) | *ppk<sup>ESB</sup>* | Zuoren Wang lab (*Guo et al., 2014*) | | https://doi.org/10.1016/j.celrep.2014.10.020 |
| Gene (*Drosophila melanogaster*) | *ΔGr32a<sup>1</sup>* | Craig Montell lab (*Moon et al., 2009*) | | https://doi.org/10.1016/j.cub.2009.07.061 |
| Chemical compound, drug | Sigmacote | Sigma Aldrich | Cat #: SLBF433V | |
| Chemical compound, drug | All-trans-retinal | Sigma Aldrich | Cat #: R2500 | 200 µM |
| Software, algorithm | Prism 7 | GraphPad Prism https://www.graphpad.com/ | RRID:SCR_002798 | |
| Software, algorithm | MATLAB 2018a | MathWorks, Natick, MA https://www.mathworks.com/products/matlab.html | RRID:SCR_006752 | |
| Software, algorithm | Fiji | NIH https://fiji.sc/ | RRID:SCR_002285 | |
| Software, algorithm | Adobe Illustrator | Adobe https://www.adobe.com/ | RRID:SCR_010279 | |
| Software, algorithm | Adobe Premiere pro | Adobe https://www.adobe.com/ | | |
| Code | Simulation of random flies/dots | This paper | | Materials and methods |
| Other | White LED/Infrared LED arrays (850 nm) | Xin Xing Yuan Guangdian https://item.taobao.com/item.htmid=20158878058 | | |

## Fly stocks

Flies were reared on a standard medium at 25°C and 60% relative humidity, under a 12:12 hr light: dark regime. Flies at 3–6 days post-eclosion were used unless otherwise indicated. *Canton S* was used as a wild-type control. $ORCO^{-/-}$, *ORCO-GAL4*, *UAS-ORCO*, $iav^{1}$, *iav-GAL4*, *nan-GAL4*, *nompC-GAL4* and $Poxn^{\Delta m22}$ were kindly provided by Yi Rao. $ppk^{ESB}$ was kindly provided by Zuoren Wang. $\Delta Gr32a^{1}$ was kindly provided by Craig Montell. *UAS-TNTE*, $Tub-GAL80^{ts}$ and *Cha-GAL80* were kindly provided by Aike Guo and Yan Li. CaLexA (LexAop-mCD8-GFP-2A-mCD8-GFP; UAS-LexA-VP16-NFAT; LexAop-GFP/Tm6B) flies were kindly provided by Jing Wang. $w^{1118}$ (BL5905), $norpA^{33}$ (BL9047), *ppk-GAL4* (BL32079), *UAS-CsChrimson* (BL55135 and BL55136), *Piezo-GAL4* (BL58771), $Piezo^{KO}$ (BL58770), $nan^{36a}$ (BL24902), $Gr64f^{-/-}$ (BL27883), *ΔPPK23* (BL33300), $Gr33a^{1}$ (BL31427) and $IR76b^{1}$ (BL51309) were obtained from the Bloomington *Drosophila* Stock Center. Mutant flies were outcrossed to *CS* background for at least eight generations.

## Social clustering assay

Social aggregation was tested using 3- to 6-day-old adult flies. Both male and female flies were tested, with the exception of some experiments (*Figure 6*, *Figure 8* and the related supplement material). One day before the experiment, flies were collected under cold anaesthesia and kept with 50 flies per vial (except otherwise noted), with females and males separate. All experiments were conducted around Zeitgeber time 1–7 in a room with 25°C and 60% humidity. Flies were allowed to habituate to the environment for 30 min before the test. The behavioural arenas were modified from glass culture dishes (internal diameter of 90 mm, unless noted otherwise) with the wall and ceiling treated with Sigmacote (Sigma-Aldrich, SLBF433V), in accordance with the product protocol, to prevent flies from walking on the side and top of the arena. The bottom of the arena was covered by 1% agar serving as a water source to keep the arena humidified, with a space of 12 mm height formed between the agar surface and the ceiling of the arena. Prior to experiments, the arenas were allowed to adjust to room temperature overnight. After quick cold anaesthesia on ice (within 1 min),

flies were carefully transferred to the centre of the arena. The arenas were back-illuminated by white LED arrays. For visual deprivation experiments, infrared LED arrays (850 nm) were used as the backlight source. The video- or time-lapse recordings started immediately when the flies were introduced to the arena; this was considered the zero hour.

Our system used a typical setting for behavioral observations but with improvements that enabled us to consistently observe social cluster formation. Besides maintaining humidity inside the arena for long-term observation, the agar pad also served as a favorable surface for the flies to walk on, while the 'slippery' coating on the side and top of the arena prevented flies from staying on these areas. Moreover, social cluster formation also requires a tightly-controlled external environment, because high temperature (>30°C), sound, vibration of the testing table, and moving shadows (of operators) could deter the clustering process. Our behavioral experiments were conducted in an isolated area within a quiet room. We also noticed that several factors, including small group size (<10 flies), mixed sex, starvation, and social isolation, negatively impacted the formation of clusters. Clustering in wild-type flies develops readily only when the internal and external perturbations discussed above are eliminated.

We estimated that over 225,000 flies of various genotypes were tested for social clustering in this project.

## Image and video acquisition and analysis

Digital cameras (Canon A720) were used to capture a sequence of raw images (8 Megapixels) of fly distribution. A custom script was written to control the camera to acquire time-lapse images for 120 min. The recording began as soon as the flies were first introduced into the arena. The captured images were imported to Matlab for further analysis with custom scripts.

For dynamic analysis experiments, videos of fly distribution in the arena were obtained with camcorders (Sony HDR-CX240E), at 50 frames per second (FPS). High-speed videos were captured by a Camera Link camera (Gazelle, FLIR) at 500 FPS and the images were saved as uncompressed files.

Details of touch-evoked encounter responses were analysed semi-automatically. Tracing of flies was aided by custom scripts/GUIs written in Matlab. Throughout this paper, the distance between two flies was measured between body centres, not between body surfaces.

## Social Space Index (SSI)

Photographs of the distribution of flies between 28 and 120 min after they were introduced into the arena were used for social space analysis. A Matlab program was written to extract the positions of flies and to quantify the social aggregation by calculating the NND for each fly. The percentages of flies with NNDs in bins of 5 mm were calculated. The SSI was then obtained by subtracting the percentage of flies in the second bin from that in the first bin (SSI = bin1 − bin2), as previously described (*Simon et al., 2012*).

## Simulation with 'random flies' and 'random dots'

### Random flies

For simulating flies without interactions between each other in arenas of similar size, we generated 'random flies' which followed the natural distribution of actual flies. A Matlab script first randomly picked digital 'flies' from the dataset of wild-type flies of corresponding gender and experimental conditions. These flies were plotted onto a digital arena (a circle with a diameter of 90 mm) using their original coordinates, body lengths and orientations. Flies that overlapped with others already existed in the arena were then removed. This process was repeated until the total number of virtual flies in the digital arena finally reached 50. The constituted arenas with 'scrambled' flies were then treated similarly to the real arenas, with these flies identified to provide controls for comparison with the actual flies. Because they came from the same dataset, the random flies shared the same overall spatial distribution of the actual flies, including frequent occurrence near the edge of arenas.

Random males and random females were generated from the dataset of wild-type males and females, respectively. Because random males and females displayed almost identical averaged properties in our analysis, only the results from random females were shown in this paper.

## Random dots

To generate a group of a given number of random dots, a Matlab script generated a series of dots with random locations within a circular arena (diameter: 90 mm), with the restriction that each dot was not in the proximity of the other dots (within an impermissible distance, or minimal allowed distance). The positions of these randomly generated dots were then used to compute their distribution. When the minimal allowed distance was 0 mm, the distribution of these random dots followed a Poisson distribution. When the minimal allowed distance was larger than 0 mm, their distribution followed a Matérn hard-core point process (type II) with this distance serving as the hard-core distance (*Turner, 2015*). To simplify the narration, we used the hard-core distance for both distributions.

Using spatial statistics to construct 'random dots' in this way enabled us to obtain spatial point processes that would not be observed otherwise in real experiments. In a 90 mm dish, random dots with a total number ranging from 5 to 100 and a hard-core distance ranging from 0 to 10 mm were generated.

## Fly identification and cluster reconstruction

Several Matlab scripts worked sequentially to automatically find and join the flies, whose surrounding areas and distances to neighbours met predefined criteria (described in the following section), into a cluster (also see *Figure 2A*; *Figure 2—video 1*).

The edge of the arena was manually or automatically labelled to generate a mask that allowed the following calculations to focus on only the inside of the arena. The background of a digital image of an arena with flies was first calculated by collecting the maximal values at each pixel position over the observed period. After background subtraction, the image was converted to a binary with a threshold. In the resultant image, groups of connected pixels were assigned as individual flies, each of which was fitted to an ellipse to obtain geometric parameters (body length, body width, body orientation, the centre of mass).

The inversion of the binary image resulted in a connected area between all flies and the edge of the arena. By repeatedly applying morphological shrink operations on the image while preserving its Euler number, the area was reduced into connected lines that separated individual flies. The region surrounded by some of these lines (serving as borders) for each enclosed fly was designated as the residing area of that fly. Two flies were considered contiguous neighbours (contacting neighbours) only when they shared a common border.

We identified contiguous flies whose residing areas were smaller than a threshold (area), and combined them into a basic cluster. The nearby flies (with larger residing areas) were further incorporated into a basic cluster if their distance to any flies in the basic cluster was shorter than a threshold (distance). The area threshold and distance threshold are described in the following section. The latter step was repeated to expand the cluster until no more flies met the distance criterion. The flies in the resultant cluster were treated equally in the following steps, regardless of whether they joined by area or distance criteria. In rare situations where two or more clusters formed in one arena, each cluster was treated in parallel.

Flies in a cluster were classified into insiders (whose residing area only bordered with residing areas of flies from the same cluster) and outsiders (whose residing area was juxtaposed to regions not belonging to flies of the same cluster). Due to variation among outsiders, we only quantified the connectivity of insiders. Additionally, only connections between contiguous neighbours were considered for calculating the numbers and lengths of these links.

## Predefined criterial sets for cluster reconstruction

First, we calculated the area of the arena, the unit area, and the unit distance with the following formulas:

$$\mathrm{Area\,of\,Arena} = \pi^{*}(\mathrm{Radius\,of\,Arena})^{2}.$$

$$\mathrm{Unit\,Area} = \mathrm{Area\,of\,Arena} / \mathrm{Total\,Number\,of\,Flies}.$$

$$\text{Unit Distance} = \text{sqrt}(4^* \text{Unit Area} / \pi)$$
$$\text{or } 2^* \text{Radius of Arena} / \text{sqrt} (\text{Total Number of Flies}).$$

Unit Area is the area per fly, assuming the area of the arena is divided evenly among all flies. Unit Distance is the diameter of a circle with an area equal to the Unit Area.

Each criterial set was based on one of five Stringent Factors (0.25, 0.302, 0.423, 0.49 and 0.723, from most stringent to most relaxed), the square roots of which were 0.5, 0.55, 0.65, 0.70 and 0.85, respectively. The thresholds for maximal allowable area and distance for each criterial set were calculated with the corresponding Stringent Factor:

$$\text{Area Threshold} = \text{Unit Area}^* \text{Stringent Factor}$$

$$\text{Distance Threshold} = \text{Unit Distance}^* \text{sqrt} (0.5^* \text{Stringent Factor})$$

A pair of Area Threshold and Distance Threshold form a criterial set; thus, five Stringent Factors resulted in five criterial settings for clustering (CSCs) (with stringency from high to low, *Figure 2B*).

## Surgical manipulation

For surgical removal of bilateral antenna, maxillary palps and arista, 3-day-old adult *Canton S* flies were used. The flies were anaesthetised on a $CO_2$ pad and specific operations were conducted with fine forceps under a stereo microscope (Leica, S6E). To minimise $CO_2$ toxicity, we operated on five flies in each batch. After the operations, the flies (50 in a group) were transferred to a standard food vial and maintained at 25°C with 60% humidity for 2 days before behavioural testing.

## Dust-induced grooming

The dust-induced grooming experiment was modified from a previous study (*Seeds et al., 2014*). A group of 50 flies was cold anaesthetised on ice and transferred to grooming chambers containing green phosphor (modified from a glass vial with a diameter of 31.8 mm and height of 80 mm). The chamber was gently shaken to uniformly coat each fly. Excess dust was removed by tapping the flies against nylon mesh before the flies were transferred to behavioural testing chambers for video recordings. Images of dusted flies in the behaviour chamber were checked at 0, 15 and 60 min under a fluorescence stereo microscope (Leica M205). Most body parts were almost clean at 15 min.

## Optogenetic stimulation

A group of 50 flies were collected within 3 days after eclosion and transferred into a vial with regular food containing 200 µM all-trans retinal (Sigma R2500). The vials were wrapped in aluminium foil for protection from light, then kept at 25°C and 60% humidity for 2–4 days. After transferral to the test arena, flies were allowed to recover for 1 min, and then stimulated with light. An array of white LEDs was used as the source of stimulation. Unless otherwise noted, light stimulation was presented continually throughout the observation period. The light intensity was 28 mW/cm$^2$, measured using a spectrometer (CCS200/M, Thorlabs).

## Inducible inactivation

To deactivate mechanosensing neurons, the tetanus toxin light chain, TNT, was expressed in the indicated GAL4 labelled neurons continuously from the embryonic stage, except for *nompC-GAL4.* As inactivation of *nompC* neurons at an early stage is lethal, TNT was expressed exclusively in the adult stage using the TARGET system (*McGuire et al., 2004*). Flies with *nompC-GAL4*, *UAS-TNT* and *Tub-GAL80$^{ts}$* were reared at 22°C and collected within 5 days of eclosion. A 2-day temperature shift to 30°C was applied to inhibit GAL80 activity, in order to induce TNT expression. Flies were collected and combined into 50 flies per vial one day prior to the test. The behavioural test was conducted at room temperature.

## Expression patterns

Dissection and staining of the central nervous system were performed as previously described, with slight modification (*Zhan et al., 2016*). Dissection of intact brains of adult female flies was performed in cold phosphate buffered saline (PBS) under stereo microscopy and fixed in 4% fresh

paraformaldehyde solution for 2 hr on ice. The tissues were then washed with PBT (0.1% Triton X-100 in 1 × PBS) five times (15 min each), blocked for 30 min with PBT containing 5% normal goat serum, and incubated with anti-nc82 antibody (1:100) in blocking buffer for 24 hr at 4°C. After washing with PBT five times, the tissues were incubated with secondary antibody (1:100) in PBT for 48 hr at 4°C. Samples were then washed with PBT three times (15 min each) before mounting.

For the imaging of expression in the peripheral systems, wings and legs were bilaterally removed from adult flies 4 days post-eclosion with forceps on a $CO_2$ plate. For imaging of legs, after fixation in 4% freshly prepared paraformaldehyde solution for 2 hr on ice, legs were washed with PBT (0.1% Triton X-100 in 1 × PBS) three times (15 min each) before mounting.

Samples were mounted in mounting medium (Vector, H-1000) under a coverslip. All of the fluorescent images were collected using a confocal microscope (Leica SP8) and processed with ImageJ (NIH).

### CaLexA measurements

Flies with various social experiences were used for CaLexA experiments. 'Single': flies raised individually in isolation in a rearing tube (diameter: 31.8 mm, height: 80 mm) after hatching; 'Group': 50 flies (female and male) raised together after hatching; 'Re-Grouped': the singly-raised flies grouped together (10 flies/vial, diameter: 21 mm) and reared for additional 30 hr prior to imaging. 'Netted': single fly housed in a netted top tube cage with food at the bottom (diameter: 12 mm). 'Netted in group': single fly raised in a net house and surrounded by a group of untouchable flies. 'Netted re-grouped': 10 single isolated flies were put together to form a new group for 24 hr prior to testing.

For CaLexA imaging, the *ppk-GAL4 >CaLexA* flies, 16 days old, were cold anaesthetised on ice for 5 min. The legs or wings were quickly removed with forceps, and mounted in mounting medium (Vector, H-1000) under a coverslip. Confocal images were acquired under a 40 × oil immersion objective lens with a confocal microscope (Leica SP8). The sum of all pixel intensities of stacks comprising the whole regions of interest (ROI) were calculated in GFP and autofluorescence channels. Average GFP signal was used for analysis.

### Statistics

Statistical analysis was performed using Prism 7 (GraphPad Software). All experiments were performed in parallel with both experimental and control genotypes. In box and whisker plots, all data points in a data set were plotted; each box includes data from the 25th to the 75th percentile, and the line within the box indicates the median. P values were determined using the unpaired two-tailed Student's *t*-tests for pairwise comparisons, one-way ANOVA with Tukey's or Dunnett's test for comparison of multiple groups. For bar graphs, the values shown were mean ± s.e.m.

### Source data files

The source data for the behavioral analyses, summary statistics, and source code are included in the source data files.

## Acknowledgements

We thank all members of the Y Zhu lab for stimulating discussions. We especially thank A Guo, L Liu, L Tao, K Zhang, ZG Han and G Li for helpful discussion. We also thank FT Ji for contributions at the early phase of this project. In addition, we are grateful to A Guo, Y-N Jan, Y Li, Y Rao, J Wang, Z Wang and W Zhang for providing flies. We are also grateful to M Huang for scientific and administrative support, YP Zhan for helpful discussions and help with the organization of the manuscript, and DK Feng for assistance with the figures.

This work was supported by NSFC grants (9163210042), Key Research Program of Frontier Sciences of Chinese Academy of Sciences (CAS, QYZDY-SSW-SMC015), CAS Interdisciplinary Innovation Team, and Bill and Melinda Gates Foundation (OPP1119434) to Y Zhu.

## Additional information

### Funding

| Funder | Grant reference number | Author |
|---|---|---|
| National Natural Science Foundation of China | 9163210042 | Yan Zhu |
| Chinese Academy of Sciences | QYZDY-SSW-SMC015 | Yan Zhu |
| Bill and Melinda Gates Foundation | OPP1119434 | Yan Zhu |

The funders had no role in study design, data collection and interpretation, or the decision to submit the work for publication.

### Author contributions

Lifen Jiang, Conceptualization, Investigation, Methodology, Data analysis, Validation; Yaxin Cheng, Shan Gao, Yincheng Zhong, Chengrui Ma, Tianyu Wang, Data analysis, Software; Yan Zhu, Conceptualization, Software, Analysis, Visualization, Methodology, Supervision, Funding acquisition

### Author ORCIDs

Lifen Jiang (ID) https://orcid.org/0000-0002-3498-9481
Tianyu Wang (ID) https://orcid.org/0000-0003-4169-8268
Yan Zhu (ID) https://orcid.org/0000-0002-9858-9129

### Decision letter and Author response

Decision letter https://doi.org/10.7554/eLife.51921.sa1
Author response https://doi.org/10.7554/eLife.51921.sa2

## Additional files

### Supplementary files

- Source code 1. randomdot_simfunc.
- Source code 2. randomfly_from_dataset.
- Transparent reporting form

### Data availability

All data generated or analysed during this study are included in the manuscript and supporting files.

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
