## [Decision Letter]

**Acceptance summary:**

The work examines the origins of collective behaviour by developing and studying the dynamics and neural basis for group formation in a paradigm that they establish and describe for *Drosophila melanogaster*. One main finding is that *Drosophila* can spontaneously assemble into a stable cluster through a mechanism that involves dyadic interactions between individual flies. By analysis of several sensory mutants in flies, the work shows that these encounters are mediated by appendage touches and that the social distance of the cluster is regulated by pickpocket-specific mechanosensory neurons. It nicely combines new organismal biology, with genetics and mathematical modelling to provide insight to how social clusters can form through simple mechanisms.

**Decision letter after peer review:**

[Editors’ note: the authors submitted for reconsideration following the decision after peer review. What follows is the decision letter after the first round of review.]

Thank you for submitting your work entitled "Emergence of social cluster by collective dyadic actions in *Drosophila*" for consideration by *eLife*. Your article has been reviewed by three peer reviewers, and the evaluation has been overseen by Mani Ramaswami as Reviewing Editor and a Senior Editor. The following individual involved in review of your submission has agreed to reveal their identity: Mario de Bono (Reviewer #2).

Our decision has been reached after consultation between the reviewers. Based on these discussions and the individual reviews below, we regret to inform you that your work will not be considered further for publication in *eLife*.

The manuscript does provide an elegant assay for social clustering of *Drosophila* in an arena, and several interesting and innovative analyses to address the underlying mechanisms. However, as elaborated in the reviews below, there is a consensus among the reviewers that it will require several additional experiments and much more work to represent progress necessary for publication in *eLife*. The reviewers however are also supportive enough to suggest that, should the authors choose to do so, then *eLife* should remain willing to examine a completely revised and resubmitted manuscript as a new submission to the journal.

*Reviewer #1:*

Zhu and colleagues focus on using fruit flies to characterize the behavioral mechanisms underlying social clustering. They document that cluster formation occurs spontaneously over the course of minutes. It is effected by a variety of parameters such as social experience, hunger and the time of day. The authors tested the contributions of different sensory modalities to clustering, and conclude that vision, olfaction (in females) and mechanosensation contribute to clustering. The work also explores the interactions of pairs of flies and the consequences on moving in or out of the cluster to result in cluster growth. They conclude that wing length affects the social space, with longer wings increasing social space. Overall, this work is interesting. However, there are many issues and deficits that need to be addressed in order for this work to be clear and convincing.

1) The whole paper is concerned with cluster formation, but it is never properly defined. For example, in examining Figure 1C, it not clear why some flies on the edges are included in the cluster and others are excluded. Also, the authors define stages of clustering (1, 2 and 3) based on the number of flies in the cluster. However, the clusters are dynamic. How do they account for some flies moving out and reversion of stage 3 to 2 or 2 to 1? Also, how do they decide which flies on the edges are included?

2) Some additional detail would be helpful to clarify how the analysis of 50 virtual flies was performed. The code should be provided. In addition, add pictures showing the distribution of the virtual flies. An important related issue concerns edge effects. It appears that the clusters form on the edges of arenas. Therefore, the effects of the edges should be included in the modeling of the virtual flies.

3) The calculation to obtain SSI as provided in the Materials and methods is not clear. The cluster index as defined in the Materials and methods is CI = Nclustered flies/Ntotal flies. Therefore, the maximum value should be a maximum of 1.0. Yet, in the figures is scored from 0 to 100.

The term “near-neighbor” is not defined. The term "encounter" is also not defined formally. How close do flies have to be for an encounter? Does an encounter require touching? Define in the main text what is meant by "inter-fly interactions."

4) The authors mentioned that they "observed frequent appendage touches between flies when analyzing the social encountering events during the process of socialclustering." Quantification of this behavior needs to be provided.

5) To test the contributions of different sensory modalities to cluster formation, the authors surgically removed appendages and used mutants. Some mutants, such as *nan* and *iav* affect locomotion and coordination. The authors cannot use mutants such as these to conclude that mechanosensation has a role in cluster formation without devising a strategy to mitigate effects of locomotion and coordination.

6) Examining the *poxn* mutant to assess whether or not there is a role for the gustatory response for clustering is too cursory. The effects of mutations that disrupt broadly required taste receptors such as *Gr66a* should be examined. Also, by eliminating the GRNs with *poxn*, both the gustatory and contact pheromone responses are affected. The authors need to discern between these possibilities.

7) To examine a contribution of vision, the authors mention that they used IR light. I presume this is indicated as "dark" conditions, which should be relabeled as IR for clarity. It is inaccurate to conclude that a defect in cluster formation exhibited by *w1118* flies supports a role for vision. The w gene encodes an ABC transporter that is expressed in many cell types. Therefore, the *w1118* mutation could affect behavior through altering any of several sensory modalities.

8) Related to the previous point regarding *w1118* is the genetic background used for the control. Throughout the paper, the authors use Canton S (CS) as the control. However, they do not indicate whether the various mutants analyzed have been outcrossed to CS. If the backgrounds of some of their lines is *w1118*, then their conclusions as to the contributions of a given mutant to cluster formation are suspect. The authors need to outcross all of their mutants to a standard control background.

9) The data need to be presented using a consistent scheme. For example, in Figure 1—figure supplement 4, CI is expressed either as bar graphs, whisker plots, or simply by a dot plot with a horizontal line, which is not defined. The type of bar and whisker plots are also not described in the legends. It is essential that the authors are consistent throughout all figures.

10) Figure 5: To study the effect of wing length, leg length and body size on social space the authors adjusted the length of the fly wing by cutting or gluing another wing onto the existing wing to extend it. Similar methods are used to shorten legs or extend body length. However, cutting wings and legs may affect locomotor activity, which is critical for social cluster formation. Moreover, there are many gustatory neurons and mechanosensory neurons on the wing margins and tarsi. Therefore, cutting the wings would affect gustation and mechanosensation, which appear to be required for cluster formation. Gluing wings or adding copper wires to the flies could stress the flies and may strongly affect their overall activity. This manipulation might also induce glooming behavior, which could impact on cluster formation. Gluing wings or wires to a fly has too many potential unintended consequences, and cannot be properly interpreted in terms of effects on wing or body length on cluster formation. The authors need to find another way to test their hypotheses.

11) Figure 6: The *ppk*-GAL4 is expressed in multiple types of neurons. Therefore, the authors use of the *ppk*-GAL4 to manipulate the activities of mechanosensor neurons is rather preliminary, and should be followed up with similar experiments targeting different types of mechanosensory neurons.

12) Add videos illustrating various behaviors.

13) There are many examples of statements in the manuscript without referring to the relevant figure. As a result, this manuscript is unnecessarily difficult to read. Some examples are:

In the third paragraph of the subsection “Spontaneous formation of orderly social cluster”, change Figure 1—figure supplement 2B to Figure 1—figure supplement 3B.

In the fourth paragraph of the aforementioned subsection it should be Figure 1G, not Figure 1F.

In the third paragraph of the subsection “Collective dyadic interactions contribute to the clustering process”, change Figure 3B to 3C.

In the first paragraph of the subsection “Cluster grows by social encountering at its border”, change the second mention of Figure 4B-F to Figure 4—figure supplement 1.

14) The authors state that "The approaching flies preferred to use their frontal legs," however, this is not shown in Figure 3 and should be added.

*Reviewer #2:*

The paper by Jiang et al. studies clustering of *Drosophila* in a 2D environment. The authors use machine vision to extract parameters for fly-fly interactions in populations of wild-type and mutant flies. Using these data they build a picture of how flies form clusters. Their data suggest pairwise interactions sustained by multisensory inputs and modifiable by experience hold the group together. The paper is well written, and the experiments described are well-executed (although see comments below).

The paper follows previous work, notably Schneider et al., 2012 and Simon et al., 2012, both referenced in the work. There is overlap with these papers, both in methodology and results. This overlap is my main concern, as it diminishes the novelty of this paper. One way to overcome this limitation is to test if selectively inhibiting *ppk* expressing neurons in the foreleg, or the wing margin, inhibits clustering. Another is to examine in more detail why mutants described in Figure 4—figure supplement 2 fail to cluster, by careful analysis of pairwise encounters.

1) Subsection “Spontaneous formation of orderly social cluster”, third paragraph. For the simulation the authors required virtual flies to not be closer to each other than 2 mm (Materials and methods, subsection “Social cluster index and social space index”, last paragraph). It is not clear to me how this can be justified, given that many real flies approach within 1 – 2 mm of each other (Figure 1—figure supplement 3C. It seems to me the modeling needs to be redone with a 1 mm minimum distance. How does reducing the minimum distance to 1 mm effect the modeling?

2) Subsection “Cluster grows by social encountering at its border”, third paragraph. Can the authors examine further why cluster formation is delayed in mutants, and why the clusters formed rapidly dispersed in these animals? This will require looking at the formation of clusters and measuring the behavior of animals following an encounter (e.g. speed).

3) Subsection “Mechanosensory neurons are necessary for establishing normal social space”. To extend the novelty of their work, can the authors pinpoint if mechanosensory neurons in the fore-leg, or wing margin, promote clustering, by using drivers that selectively target these neurons?

*Reviewer #3:*

This manuscript addresses the baseline clustering properties of adult fruit flies. This is an area of rapid advancement and so any work here is well placed in time. The authors examine self-assembling clusters in terms of average approach distance and then use this metric to examine various sensory mutants. They conclude that ppx neurons, neurons normally associated with pain, are important. Finally, they conduct a set of prosthetic experiments, shortening or extending wings to conclude that wings size is important for social distance. There are a number of places where this work has some strengths, the big issue is that this is in fact very similar to a better conducted work published in 2012 in PNAS (Schneider et al., 2012, referenced in paper). The figures and conclusions are very similar but Schneider et al. conduct a more sophisticated analysis and arrive at more sophisticated conclusions. The authors really need to address head-on why their work is an extension of this previously published work. I am therefore not very favorable for this manuscript to be published in *eLife*.

[Editors’ note: further revisions were suggested prior to acceptance, as described below.]

Thank you for submitting your article "Emergence of social cluster by collective pairwise encounters in *Drosophila*" for consideration by *eLife*. Your article has been reviewed by two peer reviewers, and the evaluation has been overseen by Mani Ramaswami as Reviewing Editor and K VijayRaghavan as the Senior Editor. The following individual involved in review of your submission has agreed to reveal their identity: Craig Montell (Reviewer #2).

The reviewers have discussed the reviews with one another and the Reviewing Editor has drafted this decision to help you prepare a revised submission.

Summary:

The work examines the origins of collective behaviour by developing and studying the dynamics and neural basis for group formation in a paradigm that they establish and describe for *Drosophila melanogaster*. One main finding is that *Drosophila* can spontaneously assemble into a stable cluster through a mechanism that involves dyadic interactions between individual flies. By analysis of several sensory mutants in flies, the work shows that these encounters are mediated by appendage touches and that the social distance of the cluster is regulated by pickpocket-specific mechanosensory neurons. It nicely combines new organismal biology, with simple genetics and mathematical modelling to provide insight to how social clusters can form through simple mechanisms.

Essential revisions:

1) The Discussion should clearly address why social clusters that form so efficiently in these studies have not been observed previously. The careful rebuttal seems to suggest that this cannot be explained by number of flies involved or in size of the arena. A major hope is that these interesting observations can be easily reproduced in other labs. A deeper engagement with possible external needs for cluster formation will be valuable to readers in the field.

2) To broaden the impact of the findings, it will be useful to discuss whether there are other examples/observations consistent with cluster building based on pair wise interactions, and also whether, in theory, more complex models be reduced to simple pairwise rules?

3) It would be useful to formally engage, perhaps using a Monte Carlo analysis, with the possibility that simple hard-core Poisson distribution of attraction and short-scale repulsion can explain these results. If it cannot, as the authors believe, then this should be demonstrated.

---

## [Author Response]

[Editors’ note: what follows is the authors’ response to the first round of review.]

Reviewer #1:[…] Overall, this work is interesting. However, there are many issues and deficits that need to be addressed in order for this work to be clear and convincing.1) The whole paper is concerned with cluster formation, but it is never properly defined. For example, in examining Figure 1C, it not clear why some flies on the edges are included in the cluster and others are excluded.

We thank the reviewer for pointing it out. In this version, we dedicated the second subtitle (“Social cluster in fruit flies is a well-structured network”) as well as Figure 2 and its supplementary figures and Video 3 to define a cluster and explore the various criteria used to identify a cluster.

Generally, deciding whether a nearby individual belongs to a cluster is rather arbitrary without predefined thresholds. We developed a six-step procedure to automatically and objectively reconstruct a cluster from an image of a group of flies (Figure 2A, Video 3). The flies were assigned to a cluster based on the distance threshold and area threshold (Figure 2B). Varying the stringency of the criterial settings of clustering (CSC), we obtained different clusters and chose an optimal setting (CSC = 2) to study the geometric properties of a cluster (Figure 2D-G).

Also, the authors define stages of clustering (1, 2 and 3) based on the number of flies in the cluster. However, the clusters are dynamic. How do they account for some flies moving out and reversion of stage 3 to 2 or 2 to 1?

We thank the reviewer for careful inspection. With time zero referring to when flies were first introduced into the arena, the time period of 520 minutes is the growing phase of a typical cluster in wild-type flies. We divided this period into 3 successive stages (stage 1, 2 and 3) with an interval of 5 minutes to conveniently describe the changing dynamic properties of the group during cluster growth. Reversion of stages would not happen due to their fixed starting and finishing time.

Additionally, clusters formed by wild-type flies always increased their size (number of flies) during the observation period (stage 1 to 3). While there were some flies leaving the cluster, more flies joined the cluster in the same time period (Figure 3A).

We now make this clear in the text (subsection “Collective dyadic interactions contribute to the clustering process”) and figure legend (Figure 3A, C).

Also, how do they decide which flies on the edges are included?

Please see the first answer on cluster identification as well. Whether a fly near a cluster belongs to that cluster was determined by its distances to the flies belonging to that cluster. If the minimal distance was less than a threshold distance, then this fly would be included in the cluster (line 189 and Figure 2A step 4; Video 3).

On a related topic, the flies belonging to a cluster were classified into insiders and outsiders (those constituting the outer edge of the cluster) (line 205 and Figure 2A step 5).

2) Some additional detail would be helpful to clarify how the analysis of 50 virtual flies was performed. The code should be provided.

We thank the reviewer for the helpful suggestions. We now include two methods to generate random flies and random dots (details are in Materials methods).

The random flies were derived from the data set of wild-type flies of the same gender. Therefore, the random flies ultimately had the same spatial distribution as the wild-type flies, when all arenas were considered together. When a virtual arena was built to have 50 random flies, flies were randomly taken from the set of wild-type flies and added to the arena with their position, shape and orientation preserved, while the fly that would overlap with others already placed in the arena was skipped. Figure 2—figure supplement 1 shows the examples of spatial distribution of wild-type flies and random flies.

The second type of simulation was with random dots. Generated via the point process, these dots randomly distributed in an arena (Figure 2—figure supplement 4). Besides the center location, each dot also has a minimal allowed distance. When this distance is non-zero, the point process is a Hard-Core Matérn point process (Type II) (Baddeley et al., 2015); when this distance is zero, the point process is a Poisson point process.

We generated the random dots for two purposes. First, the point process allowed us to generate various spatial distributions that would not be observed in wild-type flies. Second, we used these distributions to address a related question by reviewer #3 – whether the spatial distribution of wild-type flies are the same or similar to that of Hard-Core Matérn point process.

We are happy to provide the source code in Matlab.

In addition, add pictures showing the distribution of the virtual flies.

A typical distribution of the random flies is provided in Figure 1—figure supplement 4D.

Typical distributions of random dots are provided in Figure 2—figure supplement 6A-D. The quantifications of random dots are in Figure 2—figure supplement 6E, F.

An important related issue concerns edge effects. It appears that the clusters form on the edges of arenas. Therefore, the effects of the edges should be included in the modeling of the virtual flies.

We appreciate the suggestion about the edge effect. The area of a circle increases quadratically as the radius increases. By this factor alone, there would be a higher chance for a cluster to appear near the edge, instead of the center, of an arena. On top of that, the thigmotaxis in *Drosophila* could also push flies to cluster near the edge.

In most analyses of this version, we compared the properties of wild-type flies with those of random flies. As described earlier, these random flies and the wild-type flies had similar overall spatial distribution, because they belonged to the same data set. Any edge effect in the wild-type flies would also persist in the random flies. Author response image 1 shows the cumulative spatial distribution of wild-type flies and random flies.

3) The calculation to obtain SSI as provided in the Materials and methods is not clear.

We thank the reviewer for pointing this out. In this version, we included details of the definition of SSI both in the text (subsection “Spontaneous formation of social clusters”, eighth paragraph) and Materials and methods and Figure 1. SSI is based on the histogram of the nearest-neighbor distances of a fly group which were binned with an increment of 5 mm (Figure 1P). For example, SSI in Figure 1Q was calculated as the difference between the first bin and the second bin in Figure 1P:

SSI = percentage of flies in bin1 – percentage of flies in bin2.

The cluster index as defined in the Materials and methods is CI = Nclustered flies/Ntotal flies. Therefore, the maximum value should be a maximum of 1.0. Yet, in the figures is scored from 0 to 100.

We thank the reviewer for careful observation. Cluster Index was defined as CI = Nclustered flies/Ntotal flies x 100%, so the maximum was 100 instead of 1. We found that CI caused confusions while it didn’t provide more information to justify the extra space for the main text, figures and figure legends. After our careful discussion, we decided to remove Cluster Index from this version.

The term “near-neighbor” is not defined.

Near-neighbors are the surrounding flies of a reference fly, and it was not defined strictly because no distance threshold was set to include or exclude the surrounding flies. In this version, when we consider the multiple near-neighbors, we first sorted, in ascending order, the distances of the reference fly to all other flies in the group. The flies having the first Nth smallest distances were called the N-th near-neighbors, and N is the count of near neighbors (N = 1 to 8) (Figure 1N, O; Figure 1—figure supplement 5, 6).

The term "encounter" is also not defined formally. How close do flies have to be for an encounter? Does an encounter require touching?

We thank the reviewer for raising this point. An encounter event between two flies has to meet two criteria: 1. The distance between them is within 1.5 body length; 2. Interactor is facing and walking toward interactee, regardless of the orientation of interactee. We have added the definition to the text (subsection “Collective dyadic interactions contribute to the clustering process”).

Touching does not necessarily accompany an encounter. However, in our study, about 94% of encounters displayed physical touches, and the description is added as well (subsection “Asymmetric interactions and stereotypic consequences of pair-wise social encounters”, second paragraph).

Define in the main text what is meant by "inter-fly interactions."

"Inter-fly interactions" indicates the interactions between flies. In this paper, interactions were almost exclusively between pairs (subsection “Collective dyadic interactions contribute to the clustering process”, last paragraph).

4) The authors mentioned that they "observed frequent appendage touches between flies when analyzing the social encountering events during the process of socialclustering." Quantification of this behavior needs to be provided.

We thank the reviewer for this helpful suggestion. We determined that about 94% of encounters were accompanied with physical touches (subsection “Asymmetric interactions and stereotypic consequences of pair-wise social encounters”, second paragraph). Further analyses on the touching sites, relative frequency and time-stamped touching events were in Figure 3D, E and Figure 3—figure supplement 1A, B.

5) To test the contributions of different sensory modalities to cluster formation, the authors surgically removed appendages and used mutants. Some mutants, such as nan and iav affect locomotion and coordination. The authors cannot use mutants such as these to conclude that mechanosensation has a role in cluster formation without devising a strategy to mitigate effects of locomotion and coordination.

We agree with the reviewer (and other reviewers) that abnormal locomotion and coordination in mutants would potentially affect social clustering. However, in the case of *iav^1^* and *nan^36a^*, the mutant flies exhibited sufficient ability of locomotion and coordination, if not better than the wild-type flies (CS) in our tests:

a) When analyzed the spontaneous walking activity, both mutants showed higher locomotion speed (average speed and maximal speed) than the wild-type flies (Figure 6—figure supplement 1A-C). Additionally, both mutants showed higher acceleration and deceleration than the wild-type flies, indicating their coordination of walking is unlikely an inhibitory factor for social events (Figure 6—figure supplement 1D-G).

b) When tested for climbing ability, our results indicated that *iav^1^*showed severe impairment in climbing (data of *iav^1^*shown in Author response image 2).

**Author response image 2. respfig2:** 

c) The number of social encounter events in either type of mutant is higher than the wild-type flies during the same period (Figure 6A), suggesting these mutants are very active.

Therefore, we conclude that the lack of social clustering in the mutants in our tests (including *ia^1^*and *nan^36a^*) is unlikely due to defects in coordination and activity.

6) Examining the poxn mutant to assess whether or not there is a role for the gustatory response for clustering is too cursory. The effects of mutations that disrupt broadly required taste receptors such as Gr66a should be examined. Also, by eliminating the GRNs with poxn, both the gustatory and contact pheromone responses are affected. The authors need to discern between these possibilities.

We appreciate the suggested experiments and have conducted these tests accordingly.

In order to differentiate the roles of gustatory and contact pheromone response in clustering, we tested several gustatory receptor mutants, including *Gr33a^1^*(bitter), *Gr64f ^-/-^*(sugar), *Ir76b^1^*(fatty acid and salt) and contact pheromone related mutants Δ*Gr32a^1^*and *ΔPPK23*. The GRN mutants failed to form social clusters (subsection “Multiple sensory modalities are required for cluster formation”, third paragraph) (Figure 5—figure supplement 1A), suggesting that the *poxn* phenotype includes at least an abnormal gustatory response. Additionally, loss of sense for contact pheromones also inhibit the clustering process (Figure 5—figure supplement 1A), suggesting that a normal sensation of gustatory cues and contact pheromones are required for social clustering. A key clue is the high frequency (94%) of appendage touches during the encounter event. We supposed that transmitting information of gustatory cues and contact pheromones via a single physical touch during encounter is more effective and therefore is evolutionarily favorable.

The *Gr66a* mutant (BL28804) suggested by the reviewer #1 was too weak to collect enough flies for testing – we need 50 flies per arena and multiple arenas to repeat. Our preliminary results showed that *Gr66a-GAL4* and *Gr33a-GAL4* have very similar projection patterns in the brain (though *Gr33a-GAL4* labels fewer neurons). Therefore, we tested *Gr33a^1^* instead.

7) To examine a contribution of vision, the authors mention that they used IR light. I presume this is indicated as "dark" conditions, which should be relabeled as IR for clarity.

We thank the reviewer for pointing this out. We used IR to label the flies in “dark” condition in the current version (Figure 5A).

It is inaccurate to conclude that a defect in cluster formation exhibited by w1118 flies supports a role for vision. The w gene encodes an ABC transporter that is expressed in many cell types. Therefore, the w1118 mutation could affect behavior through altering any of several sensory modalities.

We agree with the reviewer for the concern. In the revision, the results of *w1118* mutants are now removed. Instead, we tested the *norpA^33^* mutants (visual defective) as well as the wild-type flies in darkness (under IR light). Results from both testes supported that vision is required to mediate the social clustering.

8) Related to the previous point regarding w1118 is the genetic background used for the control. Throughout the paper, the authors use Canton S (CS) as the control. However, they do not indicate whether the various mutants analyzed have been outcrossed to CS. If the backgrounds of some of their lines is w1118, then their conclusions as to the contributions of a given mutant to cluster formation are suspect. The authors need to outcross all of their mutants to a standard control background.

We noticed that genetic backgrounds influence fly’s behavior. In this study, the mutant flies were backcrossed to *CS* background for 8-10 generations, regardless of their original genetic background. This description is now added to Materials and methods.

9) The data need to be presented using a consistent scheme. For example, in Figure 1—figure supplement 4, CI is expressed either as bar graphs, whisker plots, or simply by a dot plot with a horizontal line, which is not defined. The type of bar and whisker plots are also not described in the legends. It is essential that the authors are consistent throughout all figures.

We thank the reviewer for this thoughtful suggestion and have made a large number of changes throughout the revision. Now we present our results in a consistent style from figure to figure. The results of CI are removed in this version.

10) Figure 5: To study the effect of wing length, leg length and body size on social space the authors adjusted the length of the fly wing by cutting or gluing another wing onto the existing wing to extend it. Similar methods are used to shorten legs or extend body length. However, cutting wings and legs may affect locomotor activity, which is critical for social cluster formation. Moreover, there are many gustatory neurons and mechanosensory neurons on the wing margins and tarsi. Therefore, cutting the wings would affect gustation and mechanosensation, which appear to be required for cluster formation. Gluing wings or adding copper wires to the flies could stress the flies and may strongly affect their overall activity. This manipulation might also induce glooming behavior, which could impact on cluster formation. Gluing wings or wires to a fly has too many potential unintended consequences, and cannot be properly interpreted in terms of effects on wing or body length on cluster formation. The authors need to find another way to test their hypotheses.

We thank the reviewer for the insight. In this revision, we remove all of the results related to surgical manipulations on wings, legs and the body dimension.

11) Figure 6: The ppk-GAL4 is expressed in multiple types of neurons. Therefore, the authors use of the ppk-GAL4 to manipulate the activities of mechanosensor neurons is rather preliminary, and should be followed up with similar experiments targeting different types of mechanosensory neurons.

We thank the reviewer for raising this point. First, the expression patterns of *ppk*-GAL4 suggested multiple regions in the brain, VNC and the peripheral appendages (leg and wing). The well-known function of the *ppk* neurons is their response to mechanical stimulation in larvae (Adams et al., 1998, Zhong et al., 2010), or in adult (Olds and Xu, 2014, Shao et al., 2019). Therefore, *ppk* neurons mediating physical touches being critical for social clustering, is our primary working hypothesis.

As suggested by the reviewer, we had performed additional experiments to forcibly activate the other mechanosensory neurons and more candidate neurons with CsChrimson (see the list below). However, none of them displayed extensive aggregation phenotype comparable with *ppk*-GAL4, suggesting the specific role of *ppk* neurons in mediating this behavior.

Additionally, our imaging results suggested that a group of *ppk* neurons on the tip of the tarsus were activated in a contact dependent manner when flies were in a social group. Although we cannot rule out other possibilities, the most straightforward conclusion is that those tarsal *ppk* neurons participated in social clustering. We tone down our description of menchanosensation in revision as we still lack direct evidence.

List of new neurons tested:

a) Broadly expressed mechanosensory neurons: R52A06-GAL4 (BL38810), R30B01-GAL4 (BL49517), R81E10-GAL4 (BL48367) (Li et al., 2016, Hampel et al., 2017).

b) Wing mechanosensory neurons: R30B01-AD (BL70175) X R31H10-DBD (BL69835), R31H10-AD (BL69917) X R34E03- DBD (BL69836) (Hampel et al., 2017).

c) Leg mechanosensory neurons: R65A11 (BL39333), R20C06 (BL48884), R55B01 (BL39100), R13E04 (BL48565), R93A02 (BL40635), R46D02 (BL50263), R27E02 (BL49222), R93D11 (BL40654), R86G01, R74B10 (BL41300), R27B07 (BL49212). R39A11 (BL50034), R39D08 (BL50047), R22A04 (BL48963), R14F12 (BL48654), R41A08 (BL50108); R86D09, R95A11 and R46H11 (Ramdya et al., 2015).

d) Unknown neurons with their projection patterns in VNC similar to the mechanosensory GAL4s above: ~80 lines (from fly Light) (Jenett et al., 2012).

12) Add videos illustrating various behaviors.

We thank the reviewer for the suggestion. We now add 3 videos to show the various behaviors:

a) a video showing progress of social clustering in wild-type flies (Video 1);

b) a high-speed video showing the encounter event and appendage touches between the pair (Video 4);

c) a video showing pairwise interactions at the cluster edge (Video 5).

Additionally, we add two videos to help the readers to understand the related quantification methods (Video 2 for Figure 1D and Video 3 for Figure 2A).

13) There are many examples of statements in the manuscript without referring to the relevant figure. As a result, this manuscript is unnecessarily difficult to read. Some examples are:In the third paragraph of the subsection “Spontaneous formation of orderly social cluster”, change Figure 1—figure supplement 2B to Figure 1—figure supplement 3B.In the fourth paragraph of the aforementioned subsection it should be Figure 1G, not Figure 1F.In the third paragraph of the subsection “Collective dyadic interactions contribute to the clustering process”, change Figure 3B to 3C.In the first paragraph of the subsection “Cluster grows by social encountering at its border”, change the second mention of Figure 4B-F to Figure 4—figure supplement 1.

We amended all of these problems in the revision.

14) The authors state that "The approaching flies preferred to use their frontal legs," however, this is not shown in Figure 3 and should be added.

We thank the reviewer for the suggestion. We added a high-speed video and a sequence of frames from the video to illustrate the frontal leg actions of a pair during an encounter process (Video 4; Figure 3H; Figure 3—figure supplement 2)

Reviewer #2:The paper by Jiang et al. studies clustering of *Drosophila* in a 2D environment. The authors use machine vision to extract parameters for fly-fly interactions in populations of wild-type and mutant flies. Using these data they build a picture of how flies form clusters. Their data suggest pairwise interactions sustained by multisensory inputs and modifiable by experience hold the group together. The paper is well written, and the experiments described are well-executed (although see comments below).The paper follows previous work, notably Schneider et al., 2012 and Simon et al., 2012, both referenced in the work. There is overlap with these papers, both in methodology and results. This overlap is my main concern, as it diminishes the novelty of this paper.

We are grateful to reviewer #2 and highly appreciate the constructive comments by reviewer #2. We would like to articulate the novelty of our results over existing literature.

We highly respect the previous work on social spacing by Simon et al., 2012 and social interaction network (SIN) by Schneider et al., 2012. Not only did we find their results relevant, but also we learned a great deal from both papers. Therefore, both versions of our paper heavily referenced their results.

Schneider et al., 2012 focused on the iterated social interactions between fly pairs in a group of 12 flies. The authors extracted the events of transient interactions in the arena to build SIN then modeled that with social network to evaluate the transmission of information. Following their example, we used “Interacteer” and “interacttee” to describe asymmetric interactions of two flies. Also we learned to simulate independent flies from the data set of actual flies from Schneider et al., 2012.

Notably, Schneider et al., 2012 reported the social structures in a group (SIN is a network composed of interactions), but did not report any social clusters (a network composed of individual flies) or aggregations (composed of flies, but without structures or regularity) -- maybe the flies did not form clusters in their setup. Therefore it is highly likely that we worked on a different social paradigm from that of Schneider et al., 2012.

Simon et al., 2012 focused on social space. The authors are the first to use Social Space Index (SSI), which was also used in our paper. Simon et al., 2012 evaluated both horizontally-placed circular arenas and vertical triangular arenas Flies in both arena showed aggregation but without characters of regularity (see Figure 2A, 2C, 2G in Simon et al., 2012). Furthermore, the vertically placed arena might skew the distribution of flies toward the top due to their negative geotaxis. Simon et al., 2012 focused on the static social space, rather than the dynamic process to get there. In analogues, if this is the same paradigm, we studied video sequences whereas they worked on the last snapshot.

Different research goal. The first and most significant difference of our work from the two previous works is the goal. We found that a group of flies formed a cluster with order and regularity. This was not observed by Schneider et al., 2012 or Simon et al., 2012. The aggregations in Simon et al., 2012 did not reveal any structure features (Figure 2C and Figure 2G in Simon et al., 2012) and, as expected, organization, order or regularity were not analyzed or discussed at all by Simon et al., 2012.

Similarly, Schneider et al., 2012 did not report observing any social aggregations or clusters. Therefore, neither Schneider et al., 2012 nor Simon et al., 2012 studied social clustering, which is the main topic of our paper. In the revision, our goal of characterizing the social clusters and investigating the underlying mechanisms for cluster formation, is indeed unique.

Different paradigms. As discussed earlier, we worked on social cluster – a well-structured social network of individual flies, which is different from either Schneider et al., 2012 or Simon et al., 2012.

Different time window – assuming the same paradigm. Schneider et al., 2012 focused on the network of social interactions within a group. As no cluster formed there, it would be equivalent to the initiation phase of social clustering in our study (0-5 min), where no cluster core has yet formed (Figure 5E-G) – even though their analysis was between 15-45 minutes after flies were introduced into the arena. On the other hand, Simon et al., 2012 studied the end results of social interactions of a group, the final social distance, not the dynamic process of aggregation. We investigated how initially dispersed flies form a cluster, covering the initiation, development and final structures of the clusters, with the time period focus of 5-20 min.

Different approaches. Both Simon et al.,2012 and Schneider et al.,2012 tracked the fly’s distributions to reach certain quantifiable parameters and then measure those parameters in mutants to evaluate their roles in the corresponding process. Our paper followed this general scheme as well.

However, besides mutants, we manipulated the activities of target neurons to evaluate the functions of related neurons. We also employed calcium image to measure the activity of neurons after social interaction. Both approaches lead to interesting findings of *ppk* neurons (Figure 7, 8).

Notably, Schneider et al., 2012 extensively used powerful network analysis algorithms to quantify the social interaction networks between 12 flies. As admirable as an algorithm can be, it would not predict the social interaction network of 50 (or even 13) flies. It could not foresee the qualitative changes when quantitative changes reach a certain limit. For example, we now know that only when the number of flies is greater than 10, clusters are readily formed (Figure 2—figure supplement 5C). Besides the number of flies, it is likely that the size of arena would also influence the key SIN parameters as they ultimately describe the information passage where the distance is rather critical. That is, we would not know how their conclusions hold in an arena of another size.

Simon et al., 2012 varied the number of flies from 10 to 40 to test the social space of the stable aggregates. Similarly, we systematically varied the number of flies and the size of the arena to make sure the conclusions are robust in a broader situations (Figure 2—figure supplement 5, 6). Additionally, a major result that pairwise interactions near the cluster edge drive cluster growth is intrinsically independent of arena size or number of flies (Figure 4).

In Schneider et al., 2012, results of the iterative network analysis reflected the patterns of reciprocal interactions between flies. Nevertheless, their analysis assumed that the SIN is essentially a time-invariant system (Supplementary Figure 4, in Schneider et al., 2012) and the parameters of social interaction networks do not change over time. Additionally, abstraction of interactions into a SIN removes important spatial information, so a SIN inherently contains less information and therefore is less powerful and less robust. In contrast, from our work, we know that as the cluster evolves, the inter-fly interactions vary over time (Figure 3 and Figure 3—figure supplement 1) and location (Figure 4 and Figure 4—figure supplement 1, 2), especially for those interactions that occurred near the cluster edge.

Notably, the speed, duration and behavioral outputs of social interactions were not encoded by the SIN in Schneider et al., 2012. In our analysis, all three of these are critical to explain the failure of cluster formation in mutants (Figure 6). We also showed that the outputs of the encounters near the cluster edge directly promote the cluster growth (Figure 4).

Different results and conclusions. As we only studied the cluster and cluster formation, it would be clear that our results and conclusion are different from both Schneider et al., 2012 and Simon et al., 2012. The subtitles of our revision are: spontaneous formation of orderly social clusters; social cluster in fruit flies is a well-structured network; collective dyadic interactions contribute to the clustering process; asymmetric interactions and stereotypic consequences of pair-wise social encounters; cluster grows by social encountering at its border; multiple sensory modalities are required for cluster formation; abnormal encounter dynamic, rather than locomotion deficits, preclude cluster formation; *ppk* specific neurons participate in establishing normal social space; social grouping elevates activity in tarsal *ppk* neurons.

We also included the same mutants tested in Schneider et al., 2012 (*iav, Orco, Poxn*) and Simon et al., 2012 (*Orco*). Although the consensus is these mutants exhibit an abnormal behavior in a group, the exact results differ between three papers, even assuming all three groups studied the same process. For example, Schneider, 2012, suggested that olfactory input was necessary for normal global efficiency (a parameter for network organization), based on male *Orco* mutants, but Simon, 2012, suggested that olfactory input was not essential for social space, based on two male *Orco* mutants. We concluded that female flies without olfaction had abnormal social distance, but male flies without olfaction were normal, based on both mutants and wild-type flies with antenna removal (Figure 5B). Additionally, the phenotype of *Orco* in female could be rescued by *Orco* expression (Figure 5B). Another example; the rate of interactions in *CS* male is about half of that in *CS* female (Figure 1D, in Schneider et al., 2012), however, we found, the social encounter frequency in *CS* male is about 2 fold of that in *CS* female (Figure 3B, C). We could not attribute such disparity to the slight difference between defining the rate of interactions by Schneider et al., 2012 and the encounter frequency by us.

We would like to propose an ultimate test. Although both Schneider et al., 2012, and Simon et al., 2012, studied social relationships of a group of flies in an enclosed two dimensional space, combining their results could not predict reliably any major conclusions in this revision, even with the assumption that all three groups worked on different aspects of the same behavior. For this, we owe the reviewers for their constructive comments.

One way to overcome this limitation is to test if selectively inhibiting ppk expressing neurons in the foreleg, or the wing margin, inhibits clustering.

We highly appreciate the constructive and thoughtful comments, which have greatly helped us to improve our work.

We took an activity imaging approach to identify the neurons that are activated by social contacts. The *ppk* neurons in tarsus, rather than in wings, showed increased activity in flies within a group (Figure 8A-C, Figure 8—figure supplement 1). Additional experiments showed that the increase in activity is not only social related, but also social contact dependent (Figure 8D, E). Therefore, our new data suggested that tarsal *ppk* neurons are involved in social clustering.

Another is to examine in more detail why mutants described in Figure 4—figure supplement 2 fail to cluster, by careful analysis of pairwise encounters.

We assessed the entire process of clustering including initiation and development of a cluster. Interestingly, the abnormality in cluster formation in mutants was evident at the stage of cluster initiation. In wild-type flies, cluster initiation began with a small cluster serving as a core for other flies to join. Although the mutant flies form transient clusters with some delay, these clusters would not last for long (Figure 5E-K). Therefore, the arenas with these mutants would never generate a mini-cluster stable enough for the future cluster to grow.

The impairment in formation of social clusters (or a stable core) in mutants might be explained by their abnormal encountering responses. We then quantified the parameters of transient encounter events including encounter frequency, encounter duration, transient encountering speed (before and after) and behavioral choice after encountering, in mutant flies (Figure 5 and Figure 6). In comparison with the wild-type flies, the mutants displayed higher encounter frequency but shorter encounter social duration (Figure 6A, B). Besides being very active, the mutants exhibited high levels of locomotion speed both before and after the encounter (Figure 6E, H). Quantifying the behavior output after encounter indicated that pairs of mutants have a higher chance of both flies moving away, thereby a high tendency to scatter from the encounter site than wild-type flies (Figure 6C, D). These dynamic properties together might contribute to the difficulty in forming a stable core in these mutants.

1) Subsection “Spontaneous formation of orderly social cluster”, third paragraph. For the simulation the authors required virtual flies to not be closer to each other than 2 mm (Materials and methods, subsection “Social cluster index and social space index”, last paragraph). It is not clear to me how this can be justified, given that many real flies approach within 1 – 2 mm of each other (Figure 1—figure supplement 3C. It seems to me the modeling needs to be redone with a 1 mm minimum distance. How does reducing the minimum distance to 1 mm effect the modeling?

We thank the reviewer for pointing it out. The distance between two flies was measured between their body centers, not between their body surfaces. We added this distinction into the main text (subsection “Spontaneous formation of social clusters”, second paragraph) and Materials and methods (subsection “Fly identification and cluster reconstruction”, third paragraph).

The reported body size of a fly is about 3mm in length, and 2 mm in width (Manning, 1999, Patterson, et al., 1943), so previously a minimal distance of 2 mm was set in the simulation with virtual flies. The wild-type flies recognized from the binary images of arenas were smaller due to background subtraction and thresholding. We had 2.72 ± 0.21 (female) or 2.54 ± 0.19 (male) in length, and 1.05 ± 0.08 (female) or 1.03 ± 0.09 (male) in width.

Inspired by this question, we have systematically explored the minimal distance between virtual flies over a broader range (reducing to zero and increasing to 10 mm) in this version as following.

We used random flies and random dots to replace the virtual flies in the previous version. When modeling with random flies (randomly picked from the data set of wild-type flies), the minimal distance between flies in a two dimensional space allowed them to touch each other but not physically overlap with each other (Figure 1 and Materials and methods). So we set a parameter that two random flies don’t occupy the same pixel(s). This simplification is sufficient because given the high-resolution of the digital cameras used for imaging the arenas, each fly occupies about 474 ± 91 (female) or 415 ± 50 (male) pixels, or each pixel corresponding to 0.069 ± 0.004 mm of the real world. The minimal distance setting here is similar to or smaller than the real world. Nevertheless, the results of simulation with random flies are similar to that of virtual flies in the previous version.

When modeling with random dots (points without size), we analyzed their spatial patterns while varying the minimal distance between them from 0 (Poisson point process) to 10 mm (Hard-Core Matérn point process) (Figure 2—figure supplement 6 and Materials and methods). The resolution in this case is much higher, as the random coordination of flies was digitally generated and encoded with double precision floating-point numbers.

With both methods, we found that varying the minimum distance did not change the results of comparisons, and all our conclusions remained the same as with the virtual flies in the previous version. Notably, the new methods of modeling enable us to identify additional properties of the groups (Figure 1 and Figure 2).

2) Subsection “Cluster grows by social encountering at its border”, third paragraph. Can the authors examine further why cluster formation is delayed in mutants, and why the clusters formed rapidly dispersed in these animals? This will require looking at the formation of clusters and measuring the behavior of animals following an encounter (e.g. speed).

We thank the reviewer for the helpful suggestions.We compared the initiation phase of a cluster in mutants and wild-type flies. Wild-type flies formed small clusters (at least 5 flies) that would last for over a minute and continue to grow into large clusters. In the mutants, however, the appearance of small clusters was delayed, and additionally, as soon as a small cluster formed, it quickly collapsed (Figure 5E-K).We found that the mutants exhibited higher encounter frequency and shorter encounter duration (Figure 6A,B), to which the abnormal initiation of clusters might be attributed.Additionally, we measured the speeds of the flies before and after an encounter. Mutants displayed larger changes in speed (larger decrease of speed before an encounter and larger increase of speed after an encounter, Figure 6E-H). We also analyzed the spontaneous walking bouts of the mutants and wild-type flies, the average and maximal walking speeds in the tested mutants were similar to or higher than wild-type flies (Figure 6-figure supplement 1A-C). Together, these results suggest that the high encounter frequency (and also possibly short encounter duration) would arise from high motility in the mutant flies.

3) Subsection “Mechanosensory neurons are necessary for establishing normal social space”. To extend the novelty of their work, can the authors pinpoint if mechanosensory neurons in the fore-leg, or wing margin, promote clustering, by using drivers that selectively target these neurons?

We thank the reviewer for the constructive suggestion. We conducted additional experiments, optogenetically activating a collection of potential Gal4 drivers (Figure 7—figure supplement 1, please also see the response to question 11 by reviewer #1), which labeled differentially mechanosensory neurons in the leg and wing margin (Figure 7—figure supplement 3). However, none of them displayed the behavioral phenotype comparable with *ppk*-GAL4 neurons, this is not surprising considering that these drivers do not overlap with the population of *ppk* neurons. Lacking additional drivers limited us from further pursuing this direction.

Instead, we switched to calcium imaging to monitor the neuronal activity to determine the neurons activated by social interactions. Our data showed that the tarsal *ppk* neurons were activated by social grouping (Figure 8A-C). Notably, this activation was dependent on direct social contact (Figure 8D-E), suggesting the *ppk* neurons in the legs are critical for social clustering.

Reviewer #3:This manuscript addresses the baseline clustering properties of adult fruit flies. This is an area of rapid advancement and so any work here is well placed in time. The authors examine self-assembling clusters in terms of average approach distance and then use this metric to examine various sensory mutants. They conclude that ppk neurons, neurons normally associated with pain, are important. Finally, they conduct a set of prosthetic experiments, shortening or extending wings to conclude that wings size is important for social distance. There are a number of places where this work has some strengths, the big issue is that this is in fact very similar to a better conducted work published in 2012 in PNAS (Schneider et al., 2012, referenced in paper). The figures and conclusions are very similar but Schneider et al. conduct a more sophisticated analysis and arrive at more sophisticated conclusions. The authors really need to address head-on why their work is an extension of this previously published work. I am therefore not very favorable for this manuscript to be published in eLife.

We thank the reviewer for the great effort in reviewing our work, and we appreciate the helpful suggestions and comments. As for major concerns, we would like to discuss point-by-point the comments underlined above in a logical order.

1) The authors really need to address head-on why their work is an extension of this previously published work.

We thank the reviewer for this question. Study of social interactions in *Drosophila* is a relevant new field with scant papers (Aike Guo, 2017). Schneider et al., 2012 is well-known for studying dynamic aspects of group interactions in *Drosophila*. Their work provided an elegant approach to model the complex interactions in a group of flies in a two dimensional space. In fact, their results were frequently referred to or compared with in this version and the previous version. However, with due respect, our work here is not an extension of this published work, for several reasons described below.

Schneider et al., 2012 focused on the network of interactions between flies, especially pairs in a circular arena. They observed the events of social interactions in the arena and modeled that into social interaction networks (SIN) to be further analyzed with network algorithms. Following their example, we used the “Interacteer” and “interacttee” scenario to describe asymmetric interactions of two flies. Also we borrowed the idea to generate independent virtual flies for the data set of actual flies, thanks to Schneider et al., 2012.

Goals. We study how the originally loosely distributed flies form a well-organized social cluster via seemingly random interactions between flies. We discovered the phenomenon of social cluster (Figure 1) and characterized its unique properties (Figure 1 and 2). In the process, to understand the formation of such cluster, we analyzed the dynamic of local inter-fly interactions and its contribution to the cluster initiation and development (Figure 3, 4, 5). Our goal of study is different from that of Schneider et al., 2012.

Paradigms. Although also quantifying the social interactions in a group, Schneider et al., 2012 did not describe a social cluster (composed of flies). It is not clear whether social clusters similar to ours actually formed in their setup or not, as there was no mention of it. Neither SIN encoded any info of distribution of flies. If it is not for cluster formation, the network of social interactions described in Schneider et al., 2012, had a different ethological significance from ours. Furthermore, if no social clusters formed in Schneider et al., 2012, then it is dubious that the social dynamic we observed in this paper is the same as that modeled by Schneider et al., 2012. If we have a different behavioral paradigm, our study of dynamics of social interactions in a group would serve a different purpose.

Results. Schneider et al., 2012, suggested that the key parameters of SIN do not evolve over time (Supplementary Figure 4, in Schneider et al., 2012). In contrast, we showed that the pairwise interactions changed through 15 minutes (stages 1-3) of cluster development (Figure 3, 4, 6). Additionally, the interactions vary over different regions of the arena (Figure 4—figure supplement 2). Notably, our results indicated that social interaction near the edge of a cluster effectively help the cluster to grow (Figure 4). We described the initiation of cluster, the tiny cluster that would be the core for future cluster growth, was blocked in mutants (Figure 5E-K). We found that mutant flies had higher numbers of social encounters in the observed period but shorter encounter duration than wild-type flies (Figure 6). We further showed that forcible activated *ppk* neurons resulted in a compressed social cluster (Figure 7), whereas the *ppk* neurons in tarsus were activated by social grouping in a contact dependent manner (Figure 8).

In summary, the work we are presenting has a different goal of study, a different paradigm with different ethological meaning, different results and conclusions from Schneider et al., 2012. None of the results mentioned in previous paragraphs were relying on previous conclusions by Schneider et al., 2012. Nor would these findings be directly predicted by Schneider et al., 2012.

Therefore, we would consider this work is not an extension of Schneider et al., 2012.

… this is in fact very similar to a better conducted work published in 2012 in PNAS (Schneider et al., 2012, referenced in paper).

In this paper, we combined multiple approaches to study the social cluster in *Drosophila*. Besides modeling, we painstakingly quantified pair-wise interactions: frequency, location, duration, behavioral output, and contribution to the clustering process in both wild-type flies and mutants. We activated and silenced various neurons to exam their roles in social clustering. Last but not least, we took calcium imaging approach to observe the activity of *ppk* neurons induced by social interactions. With our highest regard to Schneider et al., 2012, we are hoping that our work in the current form is not of inferior quality.

Notably, an algorithm could not guarantee that when the size of arena (diameter of 60 mm) changed or even the number of the group (12 flies) changed, the model in Schneider et al., 2012, would still hold. Since only experiments would tell, we measured the cluster formation under different conditions: group size (varying the number of flies from for 5 to 100 in an arena with a diameter of 90mm), arena size (varying diameter of the arena from 90 mm to 170 mm for 50 flies) and compared the data from real flies with simulation of random dots (with hard-core distance ranging from 0 to 10 mm). Additionally, we quantified the influence of clustering by hunger status, age, and social experience.

As discussed earlier, the ethological significance of the social network in Schneider et al., 2012, is unknown. It is at least not related to social clustering, as they did not describe or report the formation of aggregations or clusters. We found it difficult to extend the conclusions in Schneider et al., 2012, to population dynamics in other social systems, such as social clustering.

The figures and conclusions are very similar but Schneider et al. conduct a more sophisticated analysis and arrive at more sophisticated conclusions

Figures. As stated above, the figures in this version are very distinct from Schneider et al., 2012. Even when it might look similar in analyzing social interactions between a pair, our purpose is to understand the underlying driving force of clustering, and the organization of figures reflects our purpose.

Conclusions. As stated above, not only our conclusions are dissimilar to Schneider et al., 2012, but our conclusions are not readily predicted from Schneider et al., 2012.

Sophistication. We have been puzzled by the word “sophisticated”. Merriam-webster defined it as 1. deprived of native or original simplicity; 2. devoid of grossness. It is our guess that here sophisticated analysis could refer to the way to generate random flies as controls or refer to designate “Interacteer” and “interacttee” to analyze the asymmetric interactions between two flies. In our revision, we also employed similar approaches (Figure 1—figure supplement 2E-H, Figure 2—figure supplement 6A-F and Figure 3D-G).

If sophisticated analysis referred to modeling the social interactions with network analysis techniques in Schneider et al., 2012, we did not conduct similar modeling as it is clear this SIN approach was too oversimplified for our purpose. Abstraction of interactions into a network based solely on the directions and numbers of encounter events of flies essentially discards important spatial and temporal information, as well as the other parameters such as speed, duration and behavioral output of each encounter event. All of these are critical to understanding abnormal clustering behavior in mutants as shown in our paper (Figure 3, 4, 6). Specifically, our analysis indicated that pairwise social interactions occurring at different locations had different effects. For example, social encounters near the cluster edge, but not in other regions of the arena, promote cluster to growth (Figure 4). Additionally, as more flies joined the cluster, the overall encounter frequency decreases over time (Figure 3C). Therefore, simply counting social events over the whole arena without taking into account the spatial and temporal variations does not fit our paradigm and analysis.

A major purpose of social network analysis is to characterize the information transmission. However, the loss of information during construction of SIN renders the SIN approach less sensitive. For example, analysis of SIN of *iav^1^* mutants with all four major parameters revealed a similar network organization as that of wild-type controls (Figure 3A-D, in Schneider et al., 2012). On the other hand, in our setup, the *iav^1^* mutants fail to form social cluster during 120 minutes (Figure 5D). Our further analysis indicated that this failure is not due to locomotion defects. Instead the mutants exhibited very high encountering frequency, short encounter duration and strong tendency to disperse away from the encounter site (Figure 6).

We took multiple simple, but effective means to quantify social interactions from different aspects, and the synthesization of these results help to reach conclusions being robust and applicable to broader settings.

In summary, Schneider et al., 2012 is an inspiring work and a solid milestone in understanding the social interactions within a group of flies. It provides tools and insights to encourage a series of future work along SIN or similar directions. However, it becoming a fixed and overgeneralized framework to gauge upcoming works is out of our expectation, as our research goal, paradigm, scope, approaches, results and conclusions are clearly different.

We do appreciate these comments which help to make this revision a better quality.

[Editors’ note: what follows is the authors’ response to the second of review.]

Essential revisions:1) The Discussion should clearly address why social clusters that form so efficiently in these studies have not been observed previously. The careful rebuttal seems to suggest that this cannot be explained by number of flies involved or in size of the arena. A major hope is that these interesting observations can be easily reproduced in other labs. A deeper engagement with possible external needs for cluster formation will be valuable to readers in the field.

We thank the editors and reviewers for this thoughtful suggestion.

In the current study, we optimized the paradigm and experimental setting to generate a relatively naturalistic environment with minimal perturbations to the flies. The details of these improvements were added to the revised Materials and methodssection as follows: “Our system used a typical setting for behavioral observations but with improvements that enabled us to consistently observe social cluster formation. […] We also noticed that several factors, including small group size (<10 flies), mixed sex, starvation, and social isolation, negatively impacted the formation of clusters. Clustering in wild-type flies develops readily only when the internal and external perturbations discussed above are eliminated”.

2) To broaden the impact of the findings, it will be useful to discuss whether there are other examples/observations consistent with cluster building based on pair wise interactions, and also whether, in theory, more complex models be reduced to simple pairwise rules?

We thank the editors and reviewers for this suggestion.

Although the detailed dynamics vary between different systems, a shared feature in collective behaviors is pairwise-based local interactions. In the Discussion section (seventh paragraph), we added a new paragraph to discuss examples of pairwise interactions in *Drosophila* and other species commonly investigated for collective behaviors.

3) It would be useful to formally engage, perhaps using a Monte Carlo analysis, with the possibility that simple hard-core Poisson distribution of attraction and short-scale repulsion can explain these results. If it cannot, as the authors believe, then this should be demonstrated.

We appreciate this important suggestion regarding modeling.

We used a static model to understand how a key feature of clustering, regular social space, is achieved by analyzing a fly’s response to the influence of other flies at different distances. In essence, a fly is attracted to walk toward others when they are far away, but is repelled by a strong repulsion when coming too close to other flies, eventually settling down at a critical distance (Figure 2—figure supplement 7).

We added these modeling results in the main text (subsection “Social clusters in fruit flies are well-structured networks”, tenth paragraph) and a new figure (Figure 2—figure supplement 7). In addition, we related our model to the other analyses in the revised Discussion section (fourth paragraph).